# DropoutTS: Sample-Adaptive Dropout for Robust Time Series Forecasting

Siru Zhong [1]   Yiqiu Liu [1]   Zhiqing Cui [1]   Zezhi Shao [2]   Fei Wang [2]   Qingsong Wen [3]   Yuxuan Liang [1]

## Abstract

Deep time series models are vulnerable to noisy data ubiquitous in real-world applications. Existing robustness strategies either prune data or rely on costly prior quantification, failing to balance effectiveness and efficiency. In this paper, we introduce **DropoutTS**, a model-agnostic plugin that shifts the paradigm from **what to learn** to **how much** to learn. DropoutTS employs a *Sample-Adaptive Dropout* mechanism: leveraging spectral sparsity to efficiently quantify instance-level noise via reconstruction residuals, it dynamically calibrates model learning capacity by mapping noise to adaptive dropout rates, selectively suppressing spurious fluctuations while preserving fine-grained fidelity. Extensive experiments across diverse noise regimes and open benchmarks show DropoutTS consistently boosts superior backbones' performance, delivering advanced robustness with negligible parameter overhead and no architectural modifications. Code is available at https://github.com/CityMind-Lab/DropoutTS.

## 1. Introduction

Time series forecasting is a pivotal task spanning climate, finance, healthcare, and industrial monitoring (Idrees et al., 2019; Karevan & Suykens, 2020; Deb et al., 2017; Zheng & Huang, 2020). The proliferation of sensor data has fueled rapid advances in deep forecasting models, with architectures based on GNNs (Wu et al., 2020; Cao et al., 2020), CNNs (Bai et al., 2018; Wu et al., 2023), MLPs (Ekambaram et al., 2023; Yi et al., 2023), and Transformers (Zhou et al., 2021; Nie et al., 2023) achieving state-of-the-art performance by capturing complex long-range dependencies.

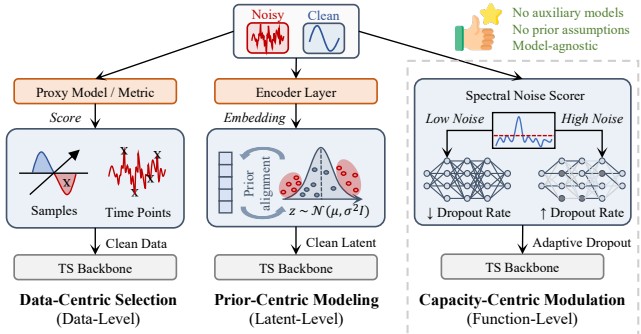

*Figure 1.* Comparison of robustness paradigms. **1) Data-Centric Selection** (Data-Level): Binary pruning causes inevitable information loss. **2) Prior-Centric Modeling** (Latent-Level): Rigid probabilistic constraints raise complexity. **3) Capacity-Centric Modulation** (Function-Level): Dynamically calibrates capacity via sample-adaptive dropout to balance fidelity and robustness.

While architectural innovations have dominated research (Kim et al., 2024; Chen et al., 2025b), the *quality* of training data remains largely ignored (Yang et al., 2025).

However, time series forecasting is inherently characterized by *sample-level heterogeneity*: different input sequences exhibit vastly different predictability due to regime shifts, varying noise levels, and temporal dynamics (Yamanishi & Takeuchi, 2002). In real-world scenarios, some samples correspond to stable, highly predictable regimes, while others are dominated by abrupt changes, rare events, or stochastic fluctuations that manifest as severe noise. Deep forecasters, characterized by high capacity, are notoriously prone to overfitting these noise samples, memorizing spurious patterns rather than learning generalizable dynamics (Lin et al., 2025). To address this mismatch, existing robustness strategies typically fall into two distinct paradigms:

- **Data-Centric Selection:** Approaches like RobustTSF (Cheng et al., 2024) and Selective Learning (Fu et al., 2025) frame robustness as a screening task. They employ auxiliary proxy models or statistical metrics to rigorously exclude samples or time steps identified as artifacts. While effective at filtering outliers, this binary "keep-or-drop" decision inevitably incurs *information loss*, risking the discarding of valid rare events that mimic noise features.

- **Prior-Centric Modeling:** Methods such as RSTIB (Chen et al., 2025a) and BayesTSF (Pan et al., 2024) disentangle noise through probabilistic frameworks. Using VAEs

[1]The Hong Kong University of Science and Technology (Guangzhou), China [2]Chinese Academy of Sciences, China [3]Squirrel Ai Learning, USA. Correspondence to: Yuxuan Liang <yuxliang@outlook.com>, Qingsong Wen <qingsongedu@gmail.com>.

*Proceedings of the 43rd International Conference on Machine Learning*, Seoul, South Korea. PMLR 306, 2026. Copyright 2026 by the author(s).

(Kingma & Welling, 2014) or BNNs (Jospin et al., 2022), they constrain latent representations to conform to specific priors (e.g., Gaussian) (Christmas & Everson, 2010). However, these *rigid distributional assumptions* fail to capture the non-stationary nature of real-world series and incur significant computational overhead.

Despite their success, both paradigms neglect the intrinsic variability of learning difficulty across samples. Consequently, the trade-off between strict data pruning and complex prior imposition fails to resolve this core challenge. Faced with such intrinsic variability, existing models adopt uniform regularization (e.g., fixed-rate dropout), implicitly assuming identical capacity requirements across samples. As illustrated in Figure 1, we propose a third paradigm: **Capacity-Centric Modulation**, rooted in the core principle of *sample-adaptive regularization*. Instead of determining a sample's usability (Selection) or fitting distribution (Modeling), we dynamically modulate model capacity allocated to each sample proportionate to its intrinsic predictability.

To this end, we introduce **DropoutTS**, a model-agnostic plugin regulating information flow via *sample-adaptive dropout*. Leveraging spectral sparsity, its core insight is that valid temporal patterns align with dominant high-energy frequencies, while noise manifests as a diffuse low-amplitude background (Donoho, 2006). We propose a differentiable spectral noise scorer that quantifies instance-level noise via reconstruction residuals from learnable spectral masking. These residuals dictate adaptive dropout rates: higher dropout for noisy samples restricts capacity (forcing the network to "skim" dominant patterns), while relaxed dropout for clean samples enables scrutiny of fine-grained details.

Crucially, DropoutTS is **orthogonal** and **complementary** rather than competitive to existing strategies. Unlike data selection (screening the *data space*) or prior modeling (constraining the *latent space*), it acts directly on the *functional space* via dynamic capacity modulation (Table 1). This orthogonality allows it to serve as a universal robustness plugin, seamlessly integrable into diverse backbones or combinable with existing strategies for synergistic robustness gains. By extracting noise signals directly from intrinsic spectral properties in a single forward pass, DropoutTS enhances robustness without auxiliary estimators or the theoretical constraints of complex probabilistic models.

Our main contributions are summarized as follows:

- **Novel Paradigm:** We introduce a *Capacity-Centric Modulation*. Unlike binary data pruning or rigid prior modeling, DropoutTS dynamically calibrates instance-wise capacity based on spectral fidelity, reconciling the information retention-robust generalization trade-off.
- **Universal Compatibility:** DropoutTS serves as a model-agnostic, proxy-free plugin quantifies intrinsic noise with spectral sparsity, integrating seamlessly into arbitrary backbones and orthogonal to existing defense strategies.
- **Superior Performance:** Rigorous evaluations on 7 real-world datasets and the controllable **Synth-12** benchmark (composite signal-noise regimes) confirm DropoutTS 's SOTA adversarial robustness, with up to 46.0% and average 9.8% gains over six SOTA backbones.

**Conflict of Interest Disclosure.** The authors declare that they have no conflict of interest.

## 2. Related Work

**Deep Models for Time Series Forecasting.** Deep learning has propelled forecasting capabilities by capturing intricate non-linear correlations. Early recurrent architectures (RNNs (Lai et al., 2018; Petneházi, 2019), LSTMs (Hua et al., 2019)) were often hindered by long-term dependency limitations. Transformers surmounted this barrier via attention mechanisms; notably, Informer (Zhou et al., 2021) and PatchTST (Nie et al., 2023) mitigated computational complexity through sparse attention and patch-based channel independence. Subsequently, Fedformer (Zhou et al., 2022b) and TimesNet (Wu et al., 2023) exploited frequency domain information for multi-scale modeling. Concurrently, MLP-based models (e.g., DLinear (Zeng et al., 2023)) demonstrated efficacy by addressing distribution shifts. Despite these architectural strides, the critical dimension of training data quality remains under-scrutinized. Most backbones implicitly presume clean inputs, lacking intrinsic robustness to instance-level noise. In low-SNR regimes (Cheng et al., 2024), these high-capacity models are prone to overfitting noise, compromising generalization.

**Robustness and Regularization Strategies.** To mitigate sample-level noise and fidelity variability, existing methods

*Table 1.* **Comparison of Robustness Paradigms.** Existing strategies act as filters (screening data or constraining priors), whereas DropoutTS acts as a modulator. This orthogonality allows DropoutTS to function as a universal plugin, offering complementary robustness.

| Paradigm | Core Philosophy | Representative Methods | Limitations / Key Traits |
|---|---|---|---|
| **Data-Centric Selection** (e.g., Hard Selection) | *"Avoid Noise"* Discard unlearnable samples | RobustTSF (Cheng et al., 2024) Selective Learning (Fu et al., 2025) | Information Loss (Aggressive pruning) Mutually exclusive with full data usage |
| **Prior-Centric Modeling** (e.g., Complex Priors) | *"Separate Noise"* Constrain latent distributions | BayesTSF (Pan et al., 2024) RSTIB (Chen et al., 2025a) | Rigid Assumptions (Over-complicated) High computational overhead |
| **Capacity-Centric Modulation** (Sample-Adaptive) | *"Coexist with Noise"* **Modulate** capacity to noise | **DropoutTS (Ours)** | **Orthogonal & Complementary** **Plug-and-Play** (Combines with above) |

fall into two retention-based paradigms: *1) Data-Centric Selection*: Treating robustness as screening, methods like RobustTSF (Cheng et al., 2024), Selective Learning (Fu et al., 2025) and FiLM (Zhou et al., 2022a) prune suspicious samples, time steps or frequency components. While suppressing outliers, this risks irreversible *information loss* of valid rare events. *2) Prior-Centric Modeling*: Approaches such as BayesTSF (Pan et al., 2024) and RSTIB (Chen et al., 2025a) constrain latent representations to predefined distributions (e.g., Gaussian) (Alcaraz & Strodthoff, 2022), but such *rigid assumptions* poorly capture real-world non-stationarities and add heavy computational overhead.

Dropout (Srivastava et al., 2014) remains a go-to regularizer, its Bayesian variants, including MC Dropout (Gal & Ghahramani, 2016), Variational Dropout (Kingma et al., 2015), and Concrete Dropout (Gal et al., 2017), offer principled ways to tune the rate. However, all of them operate at the *global or layer-wise* level, treating every sample as equally difficult. SynEVO (Liu et al., 2025) takes a step further by adapting dropout at the model level for cross-domain transfer, yet the same uniformity assumption persists. What is missing is *sample-level* adaptability: the ability to adjust capacity on a per-instance basis in response to intrinsic noise variability. This is precisely what **DropoutTS** provides through spectral-fidelity-driven, instance-wise dropout calibration.

## 3. Spectral Sparsity as Inductive Bias

### 3.1. Problem Formulation

Let $\mathcal{X} = \{\mathbf{x}_i\}_{i=1}^N$ denote a time series dataset where $\mathbf{x}_i \in \mathbb{R}^{L \times C}$ is a look-back window of length $L$ with $C$ channels. We aim to learn a forecaster $f_\theta : \mathbb{R}^{L \times C} \to \mathbb{R}^{H \times C}$ for a future horizon $H$. In practice, $\mathbf{x}_i$ is often an additive mixture of a clean signal $\mathbf{x}_i^\star$ and non-stationary noise $\epsilon_i$, i.e., $\mathbf{x}_i = \mathbf{x}_i^\star + \epsilon_i$. For robustness benchmarking, these components are categorized as shown in Figure 2. Specifically, the **clean signal** $\mathbf{x}_i^\star$ is defined via four fundamental dynamics: stationary (*Periodic*) and non-stationary in mean (*Trend*), frequency (*Chirp*), and variance (*AM*). The **noise profile** $\epsilon_i$ is modeled under three regimes: aleatoric uncertainty (*Gaussian*), epistemic anomalies (*Heavy-tail*), and observation failures (*Missing Values*). Standard training minimizes the empirical risk $\sum_i \ell(f_\theta(\mathbf{x}_i), \mathbf{y}_i)$, but deep models tend to overfit $\epsilon_i$ due to excessive capacity. Our goal is to dynamically regularize model capacity per sample $\mathbf{x}_i$ based on its noise level. Further details are provided in Table 12.

### 3.2. Empirical Verification of Sparsity

DropoutTS is grounded in the premise that valid temporal signals exhibit intrinsic **spectral sparsity**. For a single instance $\mathbf{x} = \mathbf{x}^\star + \epsilon$, Fourier Transform linearity implies $\mathcal{F}(\mathbf{x}) = \mathcal{F}(\mathbf{x}^\star) + \mathcal{F}(\epsilon)$; our core observation is that signal

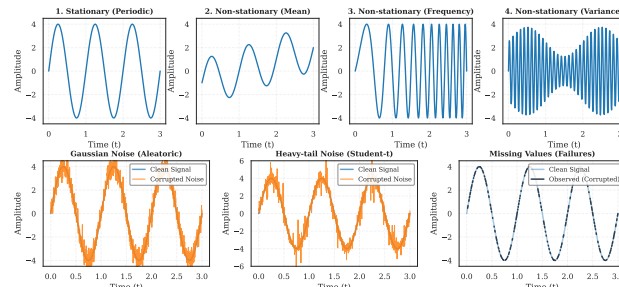

*Figure 2.* **Signal Regimes and Noise Profiles. Top:** Clean signal ranging from stationary to non-stationarity in mean (Trend), frequency (Chirp), and variance (AM). **Bottom:** Noise profiles modeling aleatoric uncertainty (Gaussian), epistemic anomalies (Heavy-tail), and observation failures (Missing Values).

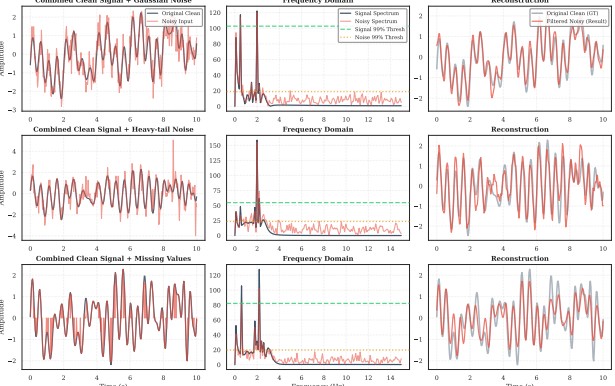

*Figure 3.* **Validation of Spectral Sparsity.** Analysis of a composite signal (Periodic + Trend + Chirp + AM) under varying noise. **Left:** Corrupted temporal inputs. **Middle:** Frequency spectra exhibit distinct separation between sparse high-energy signal and dispersed noise. **Right:** Spectral thresholding reconstruction matches ground truth, verifying robustness to outliers/missing data.

energy $|\mathcal{F}(\mathbf{x}^\star)|$ concentrates in sparse dominant frequencies, while noise $|\mathcal{F}(\epsilon)|$ manifests as a dense, low-amplitude broadband background. To validate this, we constructed a test set covering the Cartesian product of the aforementioned signal dynamics and noise profiles, and subjected samples to strict spectral truncation (retaining only top **1%** of components, $\tau$ at the $99^{th}$ percentile) to rigorously evaluate the information necessity lower bound. Figure 3 visualizes the recovery of a composite signal integrating all regimes (see Appendix C for details). We highlight two critical findings:

**1) Sparsity under Extreme Compression.** High-fidelity reconstruction verifies the invariance of spectral sparsity: valid signal energy consistently aggregates into sparse frequency-domain peaks (distinct from diffuse noise) across all complexity levels. This sparsity endows intrinsic outlier immunity (temporal spikes map to filterable broadband noise) and implicit imputation capability (dominant harmonics enable global continuity-based missing interval completion).

**2) Universality in Real-World Data.** We extended this validation on 7 real-world benchmarks (e.g., ETT) to eliminate

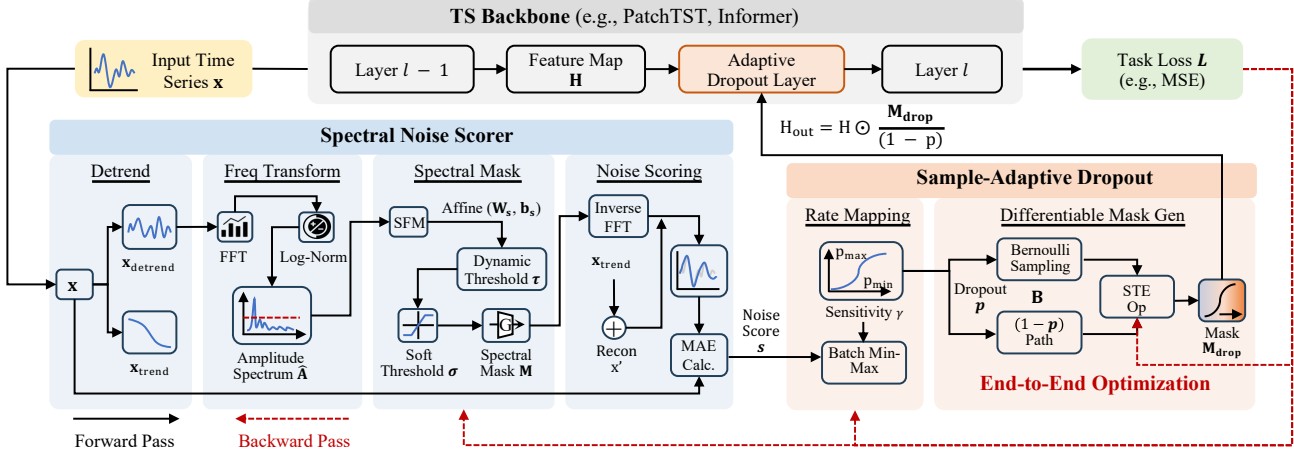

*Figure 4.* **Overview of DropoutTS.** A model-agnostic plugin replacing default dropout with adaptive dropout, consisting of two components: (1) **Spectral Noise Scorer**, which quantifies instance-level noise via spectral residual reconstruction; (2) **Sample-Adaptive Dropout**, which maps noise scores to dynamic dropout rates and generates differentiable masks for end-to-end learning.

synthetic bias. Even with stringent 90% spectral pruning, reconstructed signals retain high fidelity (Appendix D), confirming spectral sparsity as a universal property of temporal dynamics independent of specific generative processes.

**Residual as a Proxy-Free Noise Metric.** The clear spectral separation implies that simple thresholding can approximate the clean signal $\hat{\mathbf{x}}$ via $\mathbf{M}$ that mask low-amplitude fluctuations ($\tau$ denotes a spectral amplitude threshold).

$$\hat{\mathbf{x}} = \mathcal{F}^{-1}(\mathbf{M} \odot \mathcal{F}(\mathbf{x})), \quad \mathbf{M}_k = \mathbb{I}(|\mathcal{F}(\mathbf{x})|_k > \tau). \quad (1)$$

Since $\hat{\mathbf{x}} \approx \mathbf{x}^\star$, the reconstruction residual $\|\mathbf{x} - \hat{\mathbf{x}}\|$ inherently quantifies noise intensity, acting as an intrinsic **proxy-free metric** for instance-wise noise $\epsilon$ (no proxy models/complex priors) and directly motivating the design of DropoutTS.

## 4. Methodology

DropoutTS shifts the focus from data pruning or prior modeling to continuous capacity modulation. Our key idea is to predict a dropout rate for each input sample based on its spectral fidelity, and apply it consistently across the forecasting backbone to regulate the learning bottleneck. As shown in Figure 4, it comprises two core modules: a **Spectral Noise Scorer** (quantifying noise intensity via spectral sparsity) and a **Sample-Adaptive Dropout** (dynamically calibrating model capacity with a differentiable mechanism).

### 4.1. Spectral Noise Scorer

To circumvent the intractability of time-domain noise detection, we exploit the inductive bias of *Spectral Sparsity* in the frequency domain. It includes the following four steps:

**Global Linear Detrending.** Real-world time series often contain unstable trends. Applying Fast Fourier Transform (FFT) directly causes spectral leakage (Gibbs phenomenon)

(McFee, 2023), where trend discontinuities at boundaries introduce high-frequency artifacts. To mitigate this, for an input instance $\mathbf{x} \in \mathbb{R}^{L \times C}$, we remove the trend using a global linear fit via Ordinary Least Squares (OLS):

$$\mathbf{x}_{detrend} = \mathbf{x} - \mathbf{x}_{trend}, \quad \text{with } \mathbf{x}_{trend} = \mathbf{T}\mathbf{w}^* + \mathbf{b}^*, \quad (2)$$

where $\mathbf{T} \in \mathbb{R}^{L \times 1}$ is the time index vector, with $\mathbf{w}^*$ and $\mathbf{b}^*$ estimated from the sequence. This global approach leverages the consensus across all time steps to extract the underlying trend and is robust to edge noise, ensuring subsequent spectral analysis focuses purely on the intrinsic spectral structure (see Appendix E for further analysis).

**Log-scale Spectral Normalization.** We then apply FFT to the detrended signal to obtain its frequency $\mathbf{Z} \in \mathbb{C}^{(\lfloor L/2 \rfloor + 1) \times C}$, and compute the amplitude spectrum $\mathbf{A}$.

$$\mathbf{Z} = \mathcal{F}(\mathbf{x}_{detrend}), \quad \mathbf{A} = |\mathbf{Z}|, \quad (3)$$

In real-world scenarios, valid signal amplitudes can span several orders of magnitude, causing weak but valid periodic signals to be easily overshadowed. To address this critical issue, we perform normalization in the logarithmic space. We first compute the log-amplitude $\mathbf{L} = \log(1 + \mathbf{A})$ and subsequently apply instance-level min-max normalization:

$$\hat{\mathbf{A}} = \frac{\mathbf{L} - \min(\mathbf{L})}{\max(\mathbf{L}) - \min(\mathbf{L}) + \epsilon}, \quad (4)$$

where $\hat{\mathbf{A}} \in [0, 1]$ represents the normalized spectral energy profile, and $\epsilon$ is a small constant for numerical stability.

**SFM-Anchored Spectral Filter.** For signal-noise separation, we introduce a learnable soft mask mechanism, using the *Spectral Flatness Measure (SFM)* as a physical anchor instead of a random initialization threshold. SFM quantifies signal "whiteness": $\approx 1$ for noise, $\approx 0$ for clean signals.

$$\text{SFM}(\mathbf{A}) = \frac{\exp\left(\frac{1}{K} \sum_k \ln \mathbf{A}_k\right)}{\frac{1}{K} \sum_k \mathbf{A}_k}, \quad (5)$$

We define a dynamic threshold $\tau$ that adapts to the global noise level: $\tau = \sigma(\mathbf{w}_s \cdot \text{SFM}(\mathbf{A}) + \mathbf{b}_s)$. The spectral mask $\mathbf{M}$ is then generated via a soft thresholding function:

$$\mathbf{M} = \sigma\left(\text{Softplus}(\alpha) \cdot (\hat{\mathbf{A}} - \tau)\right). \qquad (6)$$

Here, $\alpha$ is a learnable sharpness parameter controlling the steepness of the cutoff, and $\sigma(\cdot)$ is the sigmoid function. $\mathbf{M}$ acts as a spectral gate: frequencies with normalized energy exceeding the dynamic threshold are retained ($\mathbf{M} \approx 1$), while noise floor components are truncated.

**Residual-based Scoring.** We reconstruct the "clean" periodic component via $\mathbf{x}'_{detrend} = \mathcal{F}^{-1}(\mathbf{Z} \odot \mathbf{M})$ and restore the trend to obtain $\mathbf{x}'$. The noise score $s$ is defined as the mean absolute error (MAE) the reconstruction residual:

$$s = \frac{1}{L \cdot C} \sum_{t,c} |\mathbf{x}_{t,c} - \mathbf{x}'_{t,c}|. \qquad (7)$$

The score $s \in \mathbb{R}^+$ serves as a proxy-free metric representing the intensity of non-sparse components (noise and outliers) that were filtered out by the spectral filter.

### 4.2. Sample-Adaptive Dropout

The second module translates the noise score $s$ into an actionable regularization strength $p$, ensuring the model "skims" noisy data while "scrutinizing" clean data.

**Batch-Aware Rate Mapping.** Since absolute noise scores vary across datasets, we normalize $s$ within each mini-batch to $\hat{s} \in [0, 1]$ using batch-wise min-max scaling. We then map this relative score to a bounded dropout probability $p \in [p_{min}, p_{max}]$ using a learnable sensitivity curve:

$$\tilde{s} = \tanh(\hat{s} \cdot \text{Softplus}(\gamma)), \qquad (8)$$

$$p = p_{min} + (p_{max} - p_{min}) \cdot \tilde{s}. \qquad (9)$$

Here, the learnable $\gamma$ modulates the sensitivity curve: clean samples ($\hat{s} \to 0$) use $p_{\min}$ dropout to preserve information, noisy samples induce $p_{\max}$ regularization.

**Differentiable Mask Generation (STE).** Standard dropout sampling involves a discrete sampling operation that is non-differentiable. To allow gradients to flow from the task loss back to the spectral filter parameters $(\alpha, \mathbf{w}_s, \mathbf{b}_s, \gamma)$, we employ the *Straight-Through Estimator (STE)* trick. During the forward pass, we sample a binary mask $\mathbf{B} \sim \text{Bernoulli}(1 - p)$. To enable backward propagation, we formulate the effective mask $\mathbf{M}_{drop}$ as:

$$\mathbf{M}_{drop} = \mathbf{B} + ((1 - p) - (1 - p)_{\text{detach}}). \qquad (10)$$

In the forward pass, the term inside the parenthesis evaluates to zero, so $\mathbf{M}_{drop} \equiv \mathbf{B}$ (discrete). In the backward pass, the

gradient ignores $\mathbf{B}$ and flows through $(1 - p)$, allowing the optimization of $p$. The final feature map $\mathbf{H}$ is modulated as:

$$\mathbf{H}_{out} = \mathbf{H} \odot \frac{\mathbf{M}_{drop}}{1 - p}. \qquad (11)$$

Theoretically, this mechanism dynamically matches model capacity to the signal's effective spectral dimension, minimizing the generalization bound (Appendix F).

**End-to-End Optimization.** Crucially, DropoutTS requires no modification to the task objective. Through the STE-enabled path, the task loss (e.g., MSE) backpropagates to optimize the noise sensitivity parameters. This creates an *implicit feedback loop*: since clean samples are inherently easier to fit, gradients drive the mechanism to relax dropout to maximize information retention, while conversely increasing regularization on noisy samples to prevent overfitting.

## 5. Experiments

In this section, we systematically verify the effectiveness, robustness, and generality of DropoutTS. We structure our analysis around four core research questions (RQ):

- **RQ1 (Robustness & Stability):** Can DropoutTS overcome static regularization and maintain consistency across clean and extremely noisy regimes? (Sec. 5.2)

- **RQ2 (Universality & Effectiveness):** Does DropoutTS, as a universal plugin, deliver consistent gains across diverse datasets and backbone architectures? (Sec. 5.3)

- **RQ3 (Few-Shot & Zero-Shot Generalization):** Does DropoutTS improve generalization under data scarcity and distribution shifts? (Sec. 5.4)

- **RQ4 (Mechanism, Efficiency & Compatibility):** How do internal components affect performance? Can DropoutTS balance training speed with inference latency and integrate with data-centric strategies? (Sec. 5.5)

### 5.1. Experimental Settings

**Datasets.** We evaluate on two benchmark categories. First, seven real-world datasets (ETTh1/h2, ETTm1/m2, Electricity, Weather, ILI) spanning energy, meteorology, and other domains. Second, to assess noise immunity under controlled conditions, we propose **Synth-12**, a physics-driven synthetic benchmark. Unlike sinusoidal baselines, it constructs composite temporal manifolds across 4 signal regimes × 3 noise types (Table 12), where "12" denotes their Cartesian product.

Two generation principles guide **Synth-12**. (1) **Dynamic Signal Coupling**: Clean signals combine non-linear trends (smoothed random walks) with quasi-periodic cycles exhibiting phase drifts and spectral shifts (Chirps/AM), testing the model's ability to track evolving frequencies. (2)

*Table 2.* Forecasting results on Synth-12 with input length 96, averaged over prediction horizons $H \in \{96, 192, 336, 720\}$. **Bold** indicates the best result, and underlining indicates the second best. The cells highlighted in gray represent our DropoutTS results (+DT). The last row reports the average relative improvement formatted as **MSE / MAE**. Complete results are provided in Table 13.

| $\sigma$ | Metric | Informer (2021) | | Crossformer (2023) | | PatchTST (2023) | | TimesNet (2023) | | iTransformer (2024) | | TimeMixer (2024) | |
|---|---|---|---|---|---|---|---|---|---|---|---|---|---|
| | | Raw | +DT | Raw | +DT | Raw | +DT | Raw | +DT | Raw | +DT | Raw | +DT |
| 0.1 | MSE | 0.966 | **0.514**(↑46.8%) | 0.450 | **0.386**(↑14.2%) | 0.542 | **0.530**(↑2.4%) | 0.902 | **0.848**(↑6.0%) | 0.523 | **0.519**(↑0.8%) | 0.545 | **0.542**(↑0.6%) |
| | MAE | 0.775 | **0.581**(↑25.0%) | 0.502 | **0.471**(↑6.2%) | 0.561 | **0.554**(↑1.3%) | 0.763 | **0.739**(↑3.2%) | 0.544 | **0.541**(↑0.6%) | 0.555 | **0.554**(↑0.2%) |
| 0.3 | MSE | 0.978 | **0.507**(↑48.2%) | 0.439 | **0.378**(↑13.9%) | 0.571 | **0.543**(↑4.9%) | 0.847 | **0.821**(↑3.1%) | 0.554 | **0.548**(↑1.1%) | 0.553 | **0.551**(↑0.4%) |
| | MAE | 0.780 | **0.577**(↑26.0%) | 0.495 | **0.464**(↑6.3%) | 0.580 | **0.564**(↑2.8%) | 0.743 | **0.730**(↑1.8%) | 0.566 | **0.560**(↑1.1%) | 0.565 | **0.561**(↑0.7%) |
| 0.5 | MSE | 0.936 | **0.494**(↑47.2%) | 0.431 | **0.391**(↑9.3%) | 0.573 | **0.556**(↑3.0%) | 0.816 | **0.794**(↑2.7%) | 0.561 | **0.554**(↑1.3%) | 0.558 | **0.553**(↑0.9%) |
| | MAE | 0.762 | **0.571**(↑25.1%) | 0.492 | **0.477**(↑3.1%) | 0.584 | **0.575**(↑1.5%) | 0.729 | **0.721**(↑1.1%) | 0.575 | **0.570**(↑0.9%) | 0.570 | **0.566**(↑0.7%) |
| 0.7 | MSE | 0.841 | **0.471**(↑44.0%) | 0.411 | **0.379**(↑7.8%) | 0.571 | **0.561**(↑1.8%) | 0.785 | **0.763**(↑2.8%) | 0.556 | **0.549**(↑1.3%) | 0.552 | **0.548**(↑0.7%) |
| | MAE | 0.726 | **0.558**(↑23.1%) | 0.481 | **0.438**(↑8.9%) | 0.584 | **0.577**(↑1.2%) | 0.721 | **0.707**(↑1.9%) | 0.578 | **0.572**(↑1.0%) | 0.567 | **0.564**(↑0.5%) |
| 0.9 | MSE | 0.828 | **0.464**(↑44.0%) | 0.409 | **0.360**(↑12.0%) | 0.552 | **0.541**(↑2.0%) | 0.742 | **0.717**(↑3.4%) | 0.535 | **0.532**(↑0.6%) | 0.536 | **0.531**(↑0.9%) |
| | MAE | 0.716 | **0.550**(↑23.2%) | 0.479 | **0.457**(↑4.6%) | 0.575 | **0.571**(↑0.7%) | 0.701 | **0.689**(↑1.7%) | 0.570 | **0.568**(↑0.4%) | 0.560 | **0.557**(↑0.5%) |
| **Avg. Impv.** | | **46.0% / 24.5%** | | **11.4% / 5.8%** | | **2.8% / 1.5%** | | **3.6% / 1.9%** | | **1.0% / 0.8%** | | **0.7% / 0.5%** | |

**Adversarial Noise Injection**: A layered corruption mechanism compounds background noise with heavy-tailed spikes and random observation failures. Varying noise intensity $\sigma \in \{0.1, 0.3, 0.5, 0.7, 0.9\}$ creates a difficulty gradient (SNR: 23.77 to 7.39 dB), stress-testing robustness against aleatoric uncertainty and epistemic anomalies (Table 3)

*Table 3.* **Statistical Profile of Synth-12.** Metrics include **SNR** (dB), **SFM**, and **MSE** (w.r.t. Ground Truth). The low baseline SFM ($\approx 0.003$) confirms the high fidelity of the uncorrupted manifold.

| Noise Level | Data Points | SNR (dB) | Clean SFM | Noisy SFM | MSE |
|---|---|---|---|---|---|
| 0.1 | 33,600 | 23.77 | 0.003 | 0.008 | 0.004 |
| 0.3 | 33,600 | 16.57 | 0.003 | 0.023 | 0.022 |
| 0.5 | 33,600 | 12.39 | 0.003 | 0.046 | 0.058 |
| 0.7 | 33,600 | 9.54 | 0.003 | 0.076 | 0.111 |
| 0.9 | 33,600 | 7.39 | 0.003 | 0.109 | 0.182 |

**Baselines.** We integrate DropoutTS into representative SOTA models: (1) *Transformer-based*: iTransformer (Liu et al., 2024), PatchTST (Nie et al., 2023), Crossformer (Zhang & Yan, 2023); (2) *MLP-based*: TimeMixer (Wang et al., 2024); (3) *CNN-based*: TimesNet (Wu et al., 2023).

**Implementation Details.** All baselines follow the same experimental setup with prediction lengths $H \in \{24, 36, 48, 60\}$ for ILI and $H \in \{96, 192, 336, 720\}$ for the other datasets, following prior works (Wu et al., 2023; Fu et al., 2025). Specifically, the input length (look-back window $L$) is fixed, with $L = 24$ for ILI and $L = 96$ for the other datasets. We utilize Adam (Kingma & Ba, 2015) for model optimization. To ensure fair evaluation, when applying DropoutTS to baseline models, we strictly follow their original hyperparameter settings and only tune the hyperparameter specific to DropoutTS (i.e. initial sensitivity ($\gamma$)). We evaluate the performance using two standard metrics: Mean Squared Error (MSE) and Mean Absolute Error (MAE). All experiments are implemented in PyTorch and conducted on 4 NVIDIA A800 80GB GPUs.

## 5.2. Robustness and The Fixed Dropout Paradox

**Setting.** A controlled stress test on Synth-12 across a noise spectrum $\sigma \in \{0.1, 0.3, 0.5, 0.7, 0.9\}$ is conducted. We compute forecasting metrics against the clean ground truth, thereby penalizing noise memorization and explicitly measuring the fidelity of the recovered signal dynamics.

**Overall Performance.** As detailed in Table 2, DropoutTS consistently enhances robustness across all noise regimes ($\sigma \in [0.1, 0.9]$). For noise-sensitive architectures, the gains are substantial: DropoutTS reduces Informer (Zhou et al., 2021)'s average MSE by **46.0%**, with peak improvements reaching **48.2%** at $\sigma = 0.3$. These benefits extend to strong SOTA baselines as well. Crossformer (Zhang & Yan, 2023) and TimesNet (Wu et al., 2023) achieve average MSE reductions of **11.4%** and **3.6%**, respectively. Even for recent robust models such as PatchTST (Nie et al., 2023), iTransformer (Liu et al., 2024), and TimeMixer (Wang et al., 2024), DropoutTS delivers consistent gains (ranging from **0.7%** to **2.8%**) without architectural modifications, confirming its broad applicability as a model-agnostic robustness enhancer.

**Mitigating the Paradox.** We analyze performance trajectories of the vulnerable Informer and robust SOTA Crossformer in Figure 5. Standard backbones exhibit irrational behavior: Informer (Figure 5a) shows a non-monotonic three-stage behavior: (i) *Low noise*: fixed dropout impairs clean feature learning; (ii) *Medium noise*: under-regularization triggers peak overfitting; (iii) *High noise*: extreme noise acts as stochastic augmentation, boosting performance. We term this the **"Fixed Dropout Paradox"**. Crossformer (Figure 5b) follows an extreme trend: error *monotonically decreases* with increasing noise. This suggests even advanced architectures suffer *regularization starvation* in clean regimes, relying on external noise for generalization. DropoutTS resolves this by dynamically calibrating regularization via spectral noise gates, lowering the error floor.

*Table 4.* Forecasting results on open benchmarks. The results are averaged over prediction lengths $H \in \{96, 192, 336, 720\}$ (for ILI, $H \in \{24, 36, 48, 60\}$). The input length is fixed at $L = 96$ (for ILI, $L = 24$). The cells highlighted in gray represent our DropoutTS results (+DT). The last row reports the average relative improvement formatted as **MSE / MAE**. Complete results see Table 14.

| Dataset | Metric | Informer (2021) | | Crossformer (2023) | | PatchTST (2023) | | TimesNet (2023) | | iTransformer (2024) | | TimeMixer (2024) | |
|---|---|---|---|---|---|---|---|---|---|---|---|---|---|
| | | Raw | +DT | Raw | +DT | Raw | +DT | Raw | +DT | Raw | +DT | Raw | +DT |
| ETTh1 | MSE | 1.337 | **1.077** (↑19.5%) | 0.449 | **0.441**(↑1.8%) | 0.459 | **0.446** (↑2.8%) | 0.534 | **0.520**(↑2.6%) | 0.447 | **0.445** (↑0.5%) | 0.458 | **0.455** (↑0.7%) |
| | MAE | 0.823 | **0.743** (↑9.7%) | 0.445 | **0.439** (↑1.4%) | 0.432 | **0.429**(↑0.7%) | 0.492 | **0.483** (↑1.8%) | 0.432 | **0.431** (↑0.2%) | 0.429 | **0.428**(↑0.2%) |
| ETTh2 | MSE | 2.657 | **1.393** (↑47.6%) | 0.795 | **0.627** (↑21.1%) | 0.384 | **0.378**(↑1.6%) | 0.480 | **0.456** (↑5.0%) | 0.384 | **0.381** (↑0.8%) | 0.386 | **0.380**(↑1.6%) |
| | MAE | 1.120 | **0.827** (↑26.2%) | 0.599 | **0.522** (↑12.9%) | 0.403 | **0.398**(↑1.2%) | 0.459 | **0.447** (↑2.6%) | 0.400 | **0.399**(↑0.3%) | 0.403 | **0.399**(↑1.0%) |
| ETTm1 | MSE | 1.479 | **1.194**(↑19.3%) | 0.413 | **0.394** (↑4.6%) | 0.396 | **0.393**(↑0.8%) | 0.519 | **0.514** (↑1.0%) | 0.400 | **0.399** (↑0.3%) | 0.393 | **0.392**(↑0.3%) |
| | MAE | 0.855 | **0.760** (↑11.1%) | 0.404 | **0.392** (↑3.0%) | 0.387 | **0.386**(↑0.3%) | 0.474 | **0.472** (↑0.4%) | 0.390 | **0.390** (0.0%) | 0.385 | **0.383**(↑0.5%) |
| ETTm2 | MSE | 3.357 | **1.634** (↑51.3%) | 0.379 | **0.351** (↑7.4%) | 0.280 | **0.280** (0.0%) | 0.348 | **0.348** (0.0%) | 0.284 | **0.284** (0.0%) | 0.278 | **0.278**(0.0%) |
| | MAE | 1.128 | **0.738** (↑34.6%) | 0.395 | **0.378** (↑4.3%) | 0.319 | **0.319** (0.0%) | 0.364 | **0.364** (0.0%) | 0.321 | **0.321** (0.0%) | 0.317 | **0.317**(0.0%) |
| Weather | MSE | 0.484 | **0.381** (↑21.3%) | 0.241 | **0.239** (↑0.8%) | 0.251 | **0.247** (↑1.6%) | 0.289 | **0.284** (↑1.7%) | 0.268 | **0.267** (↑0.4%) | 0.283 | **0.244**(↑13.8%) |
| | MAE | 0.432 | **0.373** (↑13.7%) | 0.268 | **0.266** (↑0.8%) | 0.270 | **0.266** (↑1.5%) | 0.305 | **0.303** (↑0.7%) | 0.280 | **0.279** (↑0.4%) | 0.302 | **0.264**(↑12.6%) |
| Electricity | MSE | 1.528 | **0.489** (↑68.0%) | 0.213 | **0.211** (↑0.9%) | 0.215 | **0.210**(↑2.3%) | 0.206 | **0.203** (↑1.5%) | 0.215 | **0.214** (↑0.5%) | 0.212 | **0.212** (0.0%) |
| | MAE | 0.989 | **0.459** (↑53.6%) | 0.290 | **0.288** (↑0.7%) | 0.291 | **0.286** (↑1.7%) | 0.304 | **0.301** (↑1.0%) | 0.291 | **0.274** (↑5.8%) | 0.282 | **0.282** (0.0%) |
| ILI | MSE | 7.130 | **6.235** (↑12.6%) | 5.035 | **4.321** (↑14.2%) | 3.322 | **3.126** (↑5.9%) | 6.045 | **4.872** (↑19.4%) | 3.163 | **2.994** (↑5.3%) | 3.182 | **3.173** (↑0.3%) |
| | MAE | 1.900 | **1.772** (↑6.7%) | 1.542 | **1.406** (↑8.8%) | 1.110 | **1.098** (↑1.1%) | 1.303 | **1.223** (↑6.1%) | 1.070 | **1.048**(↑2.1%) | 1.149 | **1.148** (↑0.1%) |
| **Avg. Impv.** | | **34.2% / 22.2%** | | **7.3% / 4.6%** | | **2.1% / 0.9%** | | **4.5% / 1.8%** | | **1.1% / 1.3%** | | **2.4% / 2.1%** | |

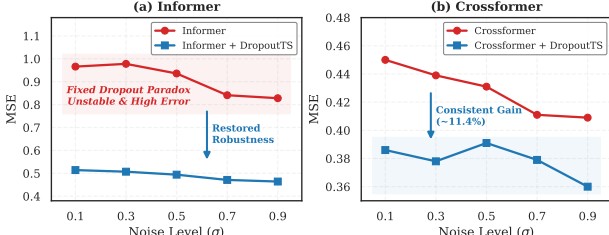

*Figure 5.* **The Fixed Dropout Paradox.** (a) Under fixed dropout, Informer produces erratic, non-monotonic error trajectories (Red). (b) Crossformer's baseline error counter-intuitively *decreases* with higher noise. DropoutTS (Blue) resolves both issues, restoring stable monotonic behavior and optimal performance on clean signals.

## 5.3. Performance on Real-world Data

**Setting.** We evaluate DropoutTS on seven widely used open benchmarks (diverse domains from energy to meteorology), integrating the same six representative backbones.

**Performance Analysis.** Table 4 compares forecasting results of various backbones with and without DropoutTS. As shown, DropoutTS serves as a universal robustness plugin, consistently boosting performance across datasets and prediction lengths regardless of Transformer-, CNN-, or MLP-based architectures. This efficacy is most visible in capacity-heavy models prone to overfitting, where DropoutTS dramatically reduces the MSE of Informer by **68.0%** on the Electricity dataset and **47.6%** on ETTh2, bridging the performance gap with modern baselines. Crucially, the method proves orthogonal to architectural advancements, squeezing out further significant gains in state-of-the-art models, such as a **13.8%** MSE reduction for TimeMixer on the Weather dataset, while also delivering clear improvements on challenging benchmarks like ILI (e.g., **19.4%** gain for TimesNet), where both data quality and sample size limit

the performance of deep models.

**Dropout Rate Analysis.** We further analyze the distribution of sample-level adaptive dropout rates ($p_i$) during final epochs to understand how DropoutTS calibrates capacity across different data regimes. As shown in Table 5, on homogeneous datasets like ETTm2, dropout rates are highly concentrated around the mean (0.329), reflecting uniform signal quality where DropoutTS defaults to near-uniform dropout. In contrast, on heterogeneous datasets like ILI, the variance is 4× larger, indicating sample diversity that demands adaptive capacity allocation. Rather than applying a blanket dropout rate, DropoutTS dynamically modulates per-sample capacity based on spectral noise characteristics.

*Table 5.* **Dropout Rate Distribution Analysis.** Adaptive dropout rates ($p_i$) at convergence. On homogeneous data (ETTm2), high concentration forces near-uniform dropout; on heterogeneous data (ILI), 4× larger variance reflects sample diversity.

| Dataset | Mean Rate | Std Dev | Variance | Interpretation |
|---|---|---|---|---|
| ETTm2 | 0.329 | 0.070 | 0.004887 | Highly concentrated (homogeneous) |
| ILI | 0.267 | 0.139 | 0.019436 | Widely dispersed (heterogeneous) |

## 5.4. Few-Shot and Zero-Shot Generalization

Table 6 (top) evaluates PatchTST on the ILI dataset with varying fractions of training data. The largest gain (**+2.85%** MAE) appears at 10% data, where models are most prone to memorizing noise due to low sample diversity. DropoutTS mitigates this by dynamically constraining functional capacity. Notably, the baseline trained on just 75% of the data matches or exceeds the 100% model, revealing significant data redundancy in the dataset; raw data scaling offers diminishing returns, highlighting the importance of data-centric innovations. For zero-shot OOD generalization, Table 6 (bottom) reports cross-dataset transfer results with

*Table 6.* **Generalization under Data Scarcity and Distribution Shifts.** Few-shot robustness (top) evaluates PatchTST with varying training fractions on ILI. Zero-shot OOD (bottom) reports cross-dataset transfer among ETT benchmarks. DropoutTS (+DT) consistently improves generalization in both regimes.

| Scenario | PatchTST (Raw) | | + DropoutTS | |
|---|---|---|---|---|
| | MAE | MSE | MAE | MSE |
| *Few-Shot Robustness on ILI* | | | | |
| 10% Data | 1.333 | 4.185 | **1.295** (↑2.85%) | **4.074** (↑2.65%) |
| 25% Data | 1.156 | 3.201 | **1.144** (↑1.04%) | **3.195** (↑0.19%) |
| 50% Data | 1.116 | 3.143 | **1.106** (↑0.90%) | **3.169** (↓0.83%) |
| 75% Data | 1.111 | 3.169 | **1.096** (↑1.35%) | **3.128** (↑1.29%) |
| 100% Data | 1.110 | 3.322 | **1.098** (↑1.08%) | **3.126** (↑5.90%) |
| **Average** | **1.165** | **3.404** | **1.148** (↑1.44%) | **3.338** (↑1.84%) |
| *Zero-Shot OOD Generalization (Cross-Dataset Transfer)* | | | | |
| ETTh1 → ETTh2 | 0.391 | 0.406 | **0.387** (↑1.02%) | **0.404** (↑0.49%) |
| ETTh1 → ETTm2 | 0.319 | 0.359 | **0.318** (↑0.31%) | **0.358** (↑0.28%) |
| ETTh2 → ETTh1 | 0.519 | 0.480 | **0.501** (↑3.47%) | **0.473** (↑1.46%) |
| ETTh2 → ETTm2 | 0.325 | 0.364 | **0.317** (↑2.46%) | **0.358** (↑1.65%) |
| ETTm1 → ETTh2 | 0.457 | 0.442 | **0.448** (↑1.97%) | **0.439** (↑0.68%) |
| ETTm1 → ETTm2 | 0.305 | 0.331 | **0.302** (↑0.98%) | **0.330** (↑0.30%) |
| ETTm2 → ETTh2 | 0.424 | 0.428 | **0.420** (↑0.94%) | **0.427** (↑0.23%) |
| ETTm2 → ETTm1 | 0.476 | 0.444 | **0.470** (↑1.26%) | **0.442** (↑0.45%) |
| **Average** | **0.402** | **0.407** | **0.395** (↑1.55%) | **0.404** (↑0.69%) |

the PatchTST backbone, covering all 8 transfer directions among ETTh1, ETTh2, ETTm1, and ETTm2. DropoutTS consistently improves zero-shot performance (e.g., ETTh2 → ETTh1: MAE $0.519 \to 0.501$), demonstrating that capacity modulation enhances robustness to distribution shifts. Baseline models tend to overfit source-domain noise and struggle under covariate shifts, while DropoutTS filters fluctuations and learns more invariant dynamics.

## 5.5. Mechanism Verification

**Ablation Study.** Table 7 quantitatively validates the contribution of each component. Removing *Global Linear Detrend* causes the most severe degradation (MSE ↓**41.7%**), confirming that non-periodic trends introduce boundary discontinuities and spectral leakage that critically mislead the scorer. Replacing the physics-informed *SFM Anchor* with a purely learnable parameter drops performance by **31.1%**, verifying spectral flatness as a necessary inductive bias for stability. Finally, excluding *Spectral Normalization* impairs generalization (↓**12.9%**) due to varying signal magnitudes, highlighting its key role in scale-invariant noise detection.

**Hyperparameter Sensitivity.** DropoutTS minimizes manual tuning by empirically fixing dropout bounds $[p_{\min}, p_{\max}] = [0.05, 0.5]$ and optimizing internal parameters end-to-end with standard initializations (mask sharpness $\alpha = 10.0$, bias $\mathbf{b}_s = 0.0$), leaving only initial sensitivity $\gamma$ as the primary tunable hyperparameter. As shown in Figure 6, the conservative setting ($\gamma = 1.0$) strikes an optimal balance, stably achieving low error across all noise levels.

*Table 7.* Ablation study of DropoutTS components on Synth-12 (averaged over $\sigma \in [0.1, 0.9]$). Baseline uses fixed dropout $p = 0.1$; Impv. denotes relative performance change vs. Baseline (↑=gain, ↓=degradation). Detailed results refer to B.3.

| Method | Components | | | MSE | | MAE | |
|---|---|---|---|---|---|---|---|
| | Detrend | Norm | SFM | Avg | Impv. | Avg | Impv. |
| Baseline | - | - | - | 1.159 | - | 0.836 | - |
| Minimal Model | None | ✗ | ✗ | 1.514 | ↓30.6% | 0.960 | ↓14.8% |
| w/o Detrend+Norm | None | ✗ | ✓ | 1.468 | ↓26.7% | 0.942 | ↓12.7% |
| w/o Detrend | None | ✓ | ✓ | 1.642 | ↓41.7% | 1.014 | ↓21.3% |
| Simple Detrend | Simple | ✓ | ✓ | 1.574 | ↓35.8% | 0.987 | ↓18.1% |
| w/o Spectral Norm | R-OLS | ✗ | ✓ | 1.308 | ↓12.9% | 0.896 | ↓7.2% |
| w/o SFM Anchor | R-OLS | ✓ | ✗ | 1.520 | ↓31.1% | 0.970 | ↓16.0% |
| **DropoutTS (Ours)** | R-OLS | ✓ | ✓ | **1.076** | ↑7.2% | **0.817** | ↑2.3% |

Notably, the aggressive strategy ($\gamma = 10.0$) outperforms under extreme noise ($\sigma \geq 0.7$), matching the need for stronger filtering in dominant noise, while the intermediate value ($\gamma = 5.0$) performs poorest. This non-monotonic behavior confirms sensitivity requires regime-specific calibration, and we recommend $\gamma = 1.0$ as a robust default.

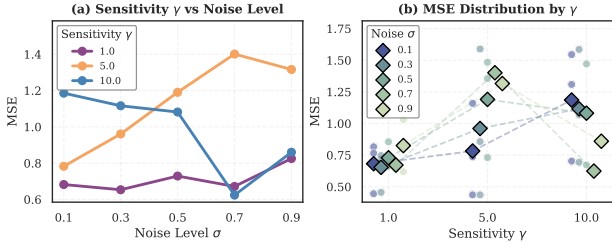

*Figure 6.* **Hyperparameter Sensitivity Analysis.** (a) Effect of $\gamma$ across noise levels ($\sigma \in [0.1, 0.9]$); (b) Error dispersion of diamonds show mean and points denotes trials.

**Efficiency Analysis.** To quantify the computational cost-performance trade-off, we conducted a stringent efficiency test on the high-noise Synth-12 benchmark under the challenging long-term forecasting (Table 8). Results confirm DropoutTS incurs negligible parameter overhead across all backbones. Although spectral operations (FFT/IFFT) add a per-epoch training latency overhead for lightweight backbones (e.g., TimeMixer, Informer). This overhead is vastly offset by *accelerated convergence*: filtering high-frequency noise that traps models in local minima, DropoutTS cuts training epochs by up to 47% (e.g., Informer), delivering a net 11% to 31% reduction in total training time. Critically, the adaptive mechanism is deactivated in evaluation, incurring zero extra inference latency and making DropoutTS well-suited for real-time deployment.

*Table 8.* **Efficiency Trade-off Analysis.** DropoutTS (+DT) incurs negligible overhead yet markedly accelerates convergence, yielding a net reduction in training budget (Raw → +DT).

| Model | Overhead (Cost) | | Benefit (Convergence) | | Net Gain | |
|---|---|---|---|---|---|---|
| | Params | Step Latency | Epochs | Total Time | Time Saved | Speedup |
| **TimeMixer** | +4 | 102.3 → 113.7 | 20 → **16** | 2045 → **1819** | **11.0%** | **1.12×** |
| **Informer** | +4 | 95.8 → 123.9 | 30 → **16** | 2873 → **1982** | **31.0%** | **1.45×** |

**Compatibility with Data-Centric Strategies.** We test whether DropoutTS competes with or complements Selective Learning (SL) (Fu et al., 2025) via combinatorial experiments on Illness (Table 9). SL reduces MSE from 8.038 to 6.461 by discarding low-quality time points. Applying DropoutTS on top of SL yields further gains, reducing MSE to **6.336** and MAE to **1.774**. The $\approx$**2%** improvement confirms orthogonality: SL screens data at the input level (data-centric), while DropoutTS modulates capacity functionally (capacity-centric), suppressing residual noise that evades binary filtering. Table 9 reports the computational efficiency of this design. DropoutTS adds only +4 parameters (0.000034% overhead) with no pre-processing, while SL requires +3,000 parameters and 29s of pre-training. Applying DropoutTS atop SL further cuts SL's training time from 15.1s to 6.5s (35.5s vs 44.1s total), making the combination both more accurate and faster to train than SL alone.

*Table 9.* **Compatibility and Efficiency Comparison.** (Top) DropoutTS complements Selective Learning on Illness ($L = 24, H = 60$). (Bottom) Computational cost on Informer+ILI.

**Compatibility with Selective Learning on Illness ($L = 24, H = 60$)**

| Method Strategy | Mechanism | Error Metric ↓ | | Rel. |
|---|---|---|---|---|
| | | **MSE** | **MAE** | **Impv** |
| **Baseline** (Raw) | - | 8.038 | 2.047 | - |
| **DropoutTS** | Capacity Modulation | 7.343 | 1.928 | 8.6% |
| **Selective Learning (SL)** | Data Selection | 6.461 | 1.902 | 19.6% |
| **SL + DropoutTS (Ours)** | **Synergistic Combo** | **6.336** | **1.774** | **21.2%** |

**Computational Cost on Informer+ILI**

| Method | Base Params | Additional | Total | Overhead |
|---|---|---|---|---|
| Raw Baseline | 11,799,047 | - | 11,799,047 | - |
| **DropoutTS (Ours)** | 11,799,047 | **+4** | **11,799,051** | **0.000034%** |
| Selective Learning | 11,799,047 | +3,000 | 11,802,047 | 0.025% |

**Training Time Comparison**

| Method | Train Time | Pre-train | Total | vs Baseline |
|---|---|---|---|---|
| Raw Baseline | 20.0s | 0s | 20.0s | - |
| **DropoutTS** | 16.8s | 0s | 16.8s | ↓15.8% |
| SL | 15.1s | 29.0s | 44.1s | ↑121.0% |
| DropoutTS + SL | 6.5s | 29.0s | 35.5s | ↑77.5% |

**Hard Denoising Comparison.** To separate destructive smoothing from adaptive capacity modulation, we compare our method against an aggressive Hard Denoising (HD) baseline that applies heavy low-pass filtering to the input signal. As shown in Table 10, HD degrades modern architectures like PatchTST and TimeMixer, confirming that blindly removing high-frequency components destroys signal structure. In contrast, DropoutTS modulates model capacity without modifying the raw data, preserving signal fidelity while suppressing noise-induced memorization.

**Qualitative Analysis.** Figure 7 shows vanilla backbones with fixed dropout often collapse into *"mean-seeking" flat lines* on noisy inputs, as standard MSE loss converges to a trivial global average under stochastic oscillations. This is a typical *over-smoothing trap* where static regularization cannot separate signal from noise. In contrast, DropoutTS acts as a spectral-guided capacity modulator: it converts

*Table 10.* **Hard Denoising vs. Capacity Modulation on ILI.** Hard Denoising (HD) aggressively smooths input signals, degrading modern architectures. In contrast, DropoutTS modulates capacity without destructively modifying the data. MAE reported.

| Backbone | Raw MAE | HD Baseline | Ours (DT) | Δ HD | Δ Ours |
|---|---|---|---|---|---|
| Informer | 1.901 | 1.869 | **1.772** | ↑1.7% | ↑**6.8%** |
| PatchTST | 1.110 | 1.162 | **1.098** | ↓4.7% | ↑**1.1%** |
| TimeMixer | 1.148 | 1.197 | **1.148** | ↓4.2% | ↑**0.5%** |
| **Overall** | - | - | - | ↓2.4% | ↑**2.8%** |

spectral noise intensity into adaptive dropout rates (instead of direct input filtering) to suppress high-frequency artifact memorization. This constraint drives the model to focus on dominant harmonics, recovering sharp, synchronized trajectories that faithfully track underlying temporal dynamics.

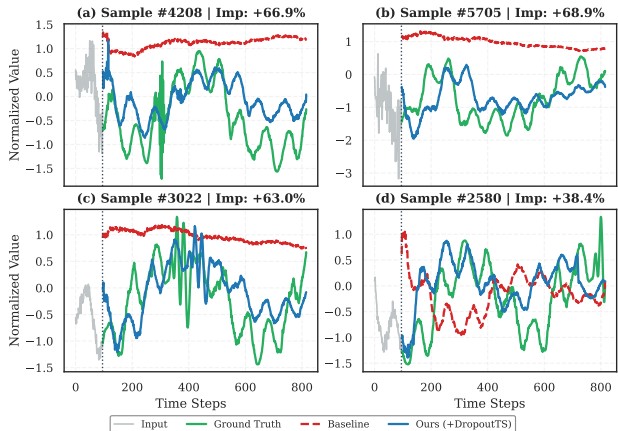

*Figure 7.* **Qualitative Visualization.** Predictive curves for DropoutTS (blue), vanilla Informer (red), and ground truth (green); sample-wise MSE improvements are shown in subtitles.

# 6. Conclusion and Future Work

This work addresses *sample-level heterogeneity* in robust forecasting through **Capacity-Centric Modulation**, moving beyond the data pruning-prior modeling dichotomy. Dynamic sample-wise dropout calibration lets DropoutTS navigate the trade-off between noise suppression and signal fidelity. We identify a core limitation of uniform regularization, the *Fixed Dropout Paradox*: static forecasters are *capacity-constrained* on clean data yet suffer *capacity overload* under noise. **Spectral sparsity** is validated as a robust, proxy-free inductive bias for intrinsic noise quantification.

**Limitations.** DropoutTS assumes valid signals concentrate in sparse dominant frequencies, failing when spectral energy is broadly distributed (e.g., random walks, chaos). Yet such processes are inherently unpredictable and lie beyond learnable forecasting. As architectural gains saturate (Wang et al., 2025), this work advances data-centric dynamic adaptation. Future work can extend capacity modulation to dimensions like depth or weight decay, and apply the spectral diagnostic to graph mining and irregularly-sampled series.

## Acknowledgements

This work is mainly supported by the Guangdong Basic and Applied Basic Research Foundation (No. 2025A1515011994). This work is also supported by the National Natural Science Foundation of China (No. 62402414, No. 62372430, No. 62502505), Guangdong Provincial Project 2025D03J0014, Guangzhou Municipal Science and Technology Project (No. 2023A03J0011), the Guangzhou Industrial Information and Intelligent Key Laboratory Project (No. 2024A03J0628), Guangdong Provincial Key Lab of Integrated Communication, Sensing and Computation for Ubiquitous Internet of Things (No. 2023B1212010007), the Youth Innovation Promotion Association CAS No.2023112, the Postdoctoral Fellowship Program of CPSF under Grant Number GZC20251078, and the China Postdoctoral Science Foundation No.2025M77154.

## Impact Statement

This paper advances time series forecasting by improving robustness to data noise. The method's efficiency and model-agnostic nature may benefit decision-making in healthcare, climate science, finance, and industrial systems. As a general machine learning technique, DropoutTS inherits standard ethical responsibilities of predictive models. The societal impact is positive when deployed responsibly.

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

# A. Experimental Details

## A.1. Benchmark Details

We conducted comprehensive evaluations on seven widely-adopted real-world benchmark datasets. As summarized in Table 11, these datasets encompass diverse domains (Energy, Meteorology, Healthcare), sampling frequencies (minutely to weekly), and time series characteristics (e.g., strong periodicity vs. irregular fluctuations). The specific details are as follows:

- **ETT (Electricity Transformer Temperature)**: Collected from two counties in China over a two-year period, this dataset tracks "oil temperature" and six power load features. It contains four subsets: *ETTh1* and *ETTh2* (hourly resolution) for long-term pattern analysis, and *ETTm1* and *ETTm2* (15-minute resolution) for fine-grained granularities.

- **Weather**: This dataset records 21 meteorological indicators (e.g., air temperature, humidity) sampled every 10 minutes throughout the year 2020. It is characterized by smooth temporal variations and strong daily periodicity.

- **Electricity**: Measurements of electric power consumption in one household with a one-minute sampling rate over 4 years. It includes various electrical quantities and sub-metering values from a house.

- **ILI (Influenza-Like Illness)**: Provided by the U.S. CDC, this dataset records the weekly ILI patient ratio from 2002 to 2021, and poses challenges with its small size, long-term temporal dependencies and seasonal outbreak patterns.

*Table 11.* Statistical characteristics of benchmark datasets. This table summarizes dimension (Dim.), total size, data split ratio, sampling frequency and application domain. ILI has distinct prediction horizons from others owing to its small sample size.

| Dataset | Dim. | Size | Split | Frequency | Prediction Length | Domain |
|---|---|---|---|---|---|---|
| ETTh1 | 7 | 14,400 | 6:2:2 | 1 hour | {96, 192, 336, 720} | Temperature |
| ETTh2 | 7 | 14,400 | 6:2:2 | 1 hour | {96, 192, 336, 720} | Temperature |
| ETTm1 | 7 | 57,600 | 6:2:2 | 15 min | {96, 192, 336, 720} | Temperature |
| ETTm2 | 7 | 57,600 | 6:2:2 | 15 min | {96, 192, 336, 720} | Temperature |
| Weather | 21 | 52,696 | 7:1:2 | 10 min | {96, 192, 336, 720} | Meteorology |
| Electricity | 321 | 26,304 | 7:1:2 | 1 hour | {96, 192, 336, 720} | Electricity |
| ILI | 7 | 966 | 7:1:2 | 1 week | {24, 36, 48, 60} | Health |

All datasets are split into train, validation, and test sets following the standard protocol established in `BasicTS` (Liang et al., 2022). This diverse testbed ensures a rigorous assessment of model robustness across different signal-to-noise regimes.

## A.2. Definition of Signal and Noise Regimes

To rigorously stress-test robustness, we formalize the definitions of clean signal dynamics ($x_t$) and noise profiles ($\tilde{x}_t$) used in our synthetic benchmark (**Synth-12**). Table 12 details the mathematical formulations and physical interpretations of these components, covering regimes from stable equilibrium to complex non-stationary shifts in mean, frequency, and variance.

*Table 12.* Detailed definitions of the signal dynamics ($x_t$) and noise corruptions ($\tilde{x}_t$). The table categorizes the fundamental components used to construct the synthetic composite manifold, mapping mathematical formulations to their physical real-world counterparts.

| Type | Category | Mathematical Formulation | Physical Interpretation | Example |
|---|---|---|---|---|
| **Signal ($x_t$)** | Stationary (Periodic) | $x_t = \sum_k A_k \sin(2\pi f_k t + \phi_k)$ | Stable equilibrium; constant freq. & amp. | Power grid voltage |
| | Non-stat. (Mean) | $x_t = \alpha t + \beta + \sum_k A_k \sin(2\pi f_k t)$ | Trend & Seasonality; drifts with fluctuations | Macroeconomic growth (GDP) |
| | Non-stat. (Freq.) | $x_t = A \sin(2\pi(f_0 t + \frac{1}{2}kt^2))$ | Spectral Drift; time-varying frequency | Doppler effects (Radar/Sonar) |
| | Non-stat. (Var.) | $x_t = (1 + \mu \sin(2\pi f_m t)) \sin(2\pi f_c t)$ | Heteroscedasticity; amplitude modulation | Financial volatility clustering |
| **Noise ($\tilde{x}_t$)** | Gaussian Noise | $\tilde{x}_t = x_t + \epsilon_t,\ \ \epsilon_t \sim \mathcal{N}(0, \sigma^2)$ | Aleatoric Uncertainty; standard noise | Sensor thermal noise |
| | Heavy-tail (Student-t) | $\tilde{x}_t = x_t + \epsilon_t,\ \ \epsilon_t \sim t_\nu\ (\nu = 2.5)$ | Epistemic Anomalies; extreme outliers | Market flash crashes |
| | Missing Values | $\tilde{x}_t = x_t \odot m_t,\ \ m_t \sim \mathcal{B}(1 - p)$ | Observation Failures; random data loss | Wireless packet loss |

# B. Complete Results

## B.1. Results of Synth-12

Table 13 presents the comprehensive forecasting performance on the **Synth-12** benchmark across the full spectrum of noise intensities ($\sigma \in [0.1, 0.9]$) and prediction horizons ($H \in \{96, \ldots, 720\}$). Observations reveal a distinct contrast in stability:

while standard backbones exhibit volatile, non-monotonic behavior as noise intensity increases, reflecting the dual nature of injected noise as both a confounding factor and an unintended regularizer, DropoutTS demonstrates *monotonic* resilience, delivering consistent MSE reductions across all noise levels without fragility to parameter drift.

Notably, in high-noise regimes ($\sigma = 0.9$), DropoutTS delivers the most substantial relative improvements (reducing Informer's average MSE by **44.0%**), validating that adaptive capacity modulation is critical when the Signal-to-Noise Ratio (SNR) is low. This robustness gap often widens at longer prediction horizons, suggesting that our spectral filtering mechanism effectively suppresses the accumulation of high-frequency noise that typically derails long-term forecasting. Furthermore, these gains are universal; while vulnerable baselines see transformative improvements, state-of-the-art models (e.g., iTransformer, TimeMixer) also exhibit consistent gains, confirming that DropoutTS provides a safety net against overfitting without compromising architectural expressivity.

*Table 13.* Full forecasting results of Synth-12 without/with DropoutTS (DT). Better results are in **bold** and underlined. The rows with gray background show the relative improvement ($\Delta$) achieved by our method (higher ↑ is better).

| Method | | Informer (2021) | | | | Crossformer (2023) | | | | PatchTST (2023) | | | | TimesNet (2023) | | | | iTransformer (2024) | | | | TimeMixer (2024) | | | |
|---|---|---|---|---|---|---|---|---|---|---|---|---|---|---|---|---|---|---|---|---|---|---|---|---|---|
| | | raw | | +DT | | raw | | +DT | | raw | | +DT | | raw | | +DT | | raw | | +DT | | raw | | +DT | |
| | Metric | MSE | MAE | MSE | MAE | MSE | MAE | MSE | MAE | MSE | MAE | MSE | MAE | MSE | MAE | MSE | MAE | MSE | MAE | MSE | MAE | MSE | MAE | MSE | MAE |
| 0.1 | 96 | 1.214 | 0.854 | **0.695** | **0.683** | 0.260 | 0.377 | **0.227** | **0.352** | 0.262 | 0.381 | **0.262** | **0.377** | 0.476 | 0.556 | **0.472** | **0.550** | 0.260 | 0.375 | **0.255** | **0.371** | 0.261 | 0.378 | **0.260** | **0.377** |
| | 192 | 0.811 | 0.722 | **0.479** | **0.561** | 0.475 | 0.511 | **0.405** | **0.478** | 0.603 | 0.595 | **0.586** | **0.589** | 0.957 | 0.797 | **0.949** | **0.791** | 0.567 | 0.564 | **0.564** | **0.563** | 0.608 | 0.590 | **0.601** | **0.587** |
| | 336 | 0.723 | 0.688 | **0.436** | **0.535** | 0.516 | 0.537 | **0.403** | **0.493** | 0.661 | 0.635 | **0.632** | **0.620** | 1.052 | 0.845 | **1.015** | **0.827** | 0.623 | 0.609 | **0.618** | **0.606** | 0.651 | 0.620 | **0.647** | **0.620** |
| | 720 | 1.114 | 0.837 | **0.445** | **0.543** | 0.549 | 0.582 | **0.507** | **0.562** | 0.644 | 0.632 | **0.635** | **0.629** | 1.122 | 0.855 | **0.957** | **0.788** | 0.642 | 0.629 | **0.639** | **0.624** | 0.659 | 0.631 | **0.659** | **0.633** |
| | Avg | 0.966 | 0.775 | **0.514** | **0.581** | 0.450 | 0.502 | **0.386** | **0.471** | 0.542 | 0.561 | **0.529** | **0.554** | 0.902 | 0.763 | **0.848** | **0.739** | 0.523 | 0.544 | **0.519** | **0.541** | 0.545 | 0.555 | **0.542** | **0.554** |
| | Improv. (Δ) | - | | 46.8%↑ | 25.0%↑ | - | | 14.2%↑ | 6.2%↑ | - | | 2.4%↑ | 1.3%↑ | - | | 6.0%↑ | 3.2%↑ | - | | 0.8%↑ | 0.6%↑ | - | | 0.6%↑ | 0.2%↑ |
| 0.3 | 96 | 1.281 | 0.875 | **0.693** | **0.679** | 0.243 | 0.373 | **0.209** | **0.334** | 0.282 | 0.401 | **0.273** | **0.393** | 0.486 | 0.565 | **0.458** | **0.545** | 0.278 | 0.392 | **0.274** | **0.388** | 0.273 | 0.391 | **0.273** | **0.387** |
| | 192 | 0.813 | 0.724 | **0.473** | **0.558** | 0.500 | 0.515 | **0.395** | **0.468** | 0.628 | 0.611 | **0.593** | **0.589** | 0.929 | 0.789 | **0.927** | **0.787** | 0.609 | 0.592 | **0.605** | **0.588** | 0.620 | 0.599 | **0.616** | **0.595** |
| | 336 | 0.718 | 0.686 | **0.436** | **0.533** | 0.513 | 0.537 | **0.430** | **0.510** | 0.709 | 0.660 | **0.649** | **0.633** | 0.994 | 0.816 | **0.966** | **0.809** | 0.661 | 0.635 | **0.652** | **0.627** | 0.660 | 0.629 | **0.655** | **0.628** |
| | 720 | 1.102 | 0.833 | **0.425** | **0.536** | 0.499 | 0.554 | **0.479** | **0.543** | 0.694 | 0.661 | **0.656** | **0.642** | 0.980 | 0.802 | **0.934** | **0.780** | 0.667 | 0.646 | **0.661** | **0.636** | 0.661 | 0.641 | **0.659** | **0.635** |
| | Avg | 0.978 | 0.780 | **0.507** | **0.577** | 0.439 | 0.495 | **0.378** | **0.464** | 0.571 | 0.580 | **0.543** | **0.564** | 0.847 | 0.743 | **0.821** | **0.730** | 0.554 | 0.566 | **0.548** | **0.560** | 0.553 | 0.565 | **0.551** | **0.561** |
| | Improv. (Δ) | - | | 48.2%↑ | 26.0%↑ | - | | 13.9%↑ | 6.3%↑ | - | | 4.9%↑ | 2.8%↑ | - | | 3.1%↑ | 1.8%↑ | - | | 1.1%↑ | 1.1%↑ | - | | 0.4%↑ | 0.7%↑ |
| 0.5 | 96 | 1.195 | 0.846 | **0.664** | **0.665** | 0.251 | 0.376 | **0.212** | **0.344** | 0.289 | 0.408 | **0.288** | **0.403** | 0.452 | 0.545 | **0.446** | **0.541** | 0.287 | 0.398 | **0.282** | **0.396** | 0.281 | 0.394 | **0.279** | **0.393** |
| | 192 | 0.782 | 0.709 | **0.446** | **0.541** | 0.518 | 0.524 | **0.411** | **0.483** | 0.624 | 0.610 | **0.618** | **0.609** | 0.897 | 0.779 | **0.874** | **0.767** | 0.619 | 0.603 | **0.612** | **0.598** | 0.628 | 0.607 | **0.620** | **0.604** |
| | 336 | 0.702 | 0.676 | **0.449** | **0.549** | 0.487 | 0.530 | **0.447** | **0.518** | 0.694 | 0.661 | **0.663** | **0.645** | 0.936 | 0.807 | **0.945** | **0.803** | 0.656 | 0.651 | **0.659** | **0.638** | 0.656 | 0.636 | **0.660** | **0.634** |
| | 720 | 1.064 | 0.819 | **0.418** | **0.528** | 0.468 | 0.537 | **0.493** | **0.562** | 0.683 | 0.656 | **0.656** | **0.643** | 0.936 | 0.783 | **0.912** | **0.772** | 0.666 | 0.649 | **0.663** | **0.648** | 0.659 | 0.641 | **0.651** | **0.632** |
| | Avg | 0.936 | 0.762 | **0.494** | **0.571** | 0.431 | 0.492 | **0.391** | **0.477** | 0.573 | 0.584 | **0.556** | **0.575** | 0.816 | 0.729 | **0.794** | **0.721** | 0.561 | 0.575 | **0.554** | **0.570** | 0.558 | 0.570 | **0.553** | **0.566** |
| | Improv. (Δ) | - | | 47.2%↑ | 25.1%↑ | - | | 9.3%↑ | 3.1%↑ | - | | 3.0%↑ | 1.5%↑ | - | | 2.7%↑ | 1.1%↑ | - | | 1.3%↑ | 0.9%↑ | - | | 0.9%↑ | 0.7%↑ |
| 0.7 | 96 | 0.943 | 0.758 | **0.624** | **0.648** | 0.245 | 0.367 | **0.206** | **0.338** | 0.298 | 0.409 | **0.287** | **0.401** | 0.476 | 0.566 | **0.444** | **0.543** | 0.290 | 0.402 | **0.285** | **0.397** | 0.288 | 0.397 | **0.280** | **0.392** |
| | 192 | 0.687 | 0.667 | **0.433** | **0.536** | 0.403 | 0.469 | **0.418** | **0.480** | 0.652 | 0.626 | **0.628** | **0.612** | 0.853 | 0.762 | **0.837** | **0.751** | 0.626 | 0.621 | **0.609** | **0.602** | 0.622 | 0.602 | **0.618** | **0.602** |
| | 336 | 0.671 | 0.661 | **0.414** | **0.526** | 0.462 | 0.515 | **0.414** | **0.497** | 0.677 | 0.646 | **0.675** | **0.652** | 0.906 | 0.785 | **0.883** | **0.773** | 0.649 | 0.648 | **0.655** | **0.647** | 0.656 | 0.634 | **0.654** | **0.634** |
| | 720 | 1.061 | 0.818 | **0.411** | **0.522** | 0.534 | 0.572 | **0.477** | **0.438** | 0.658 | 0.646 | **0.653** | **0.642** | 0.904 | 0.770 | **0.887** | **0.761** | 0.649 | 0.642 | **0.647** | **0.641** | 0.643 | 0.634 | **0.639** | **0.629** |
| | Avg | 0.841 | 0.726 | **0.471** | **0.558** | 0.411 | 0.481 | **0.379** | **0.438** | 0.571 | 0.584 | **0.561** | **0.577** | 0.785 | 0.721 | **0.763** | **0.707** | 0.556 | 0.578 | **0.549** | **0.572** | 0.552 | 0.567 | **0.548** | **0.564** |
| | Improv. (Δ) | - | | 44.0%↑ | 23.1%↑ | - | | 7.8%↑ | 8.9%↑ | - | | 1.8%↑ | 1.2%↑ | - | | 2.8%↑ | 1.9%↑ | - | | 1.3%↑ | 1.0%↑ | - | | 0.7%↑ | 0.5%↑ |
| 0.9 | 96 | 1.093 | 0.808 | **0.617** | **0.640** | 0.240 | 0.361 | **0.200** | **0.335** | 0.286 | 0.401 | **0.281** | **0.400** | 0.453 | 0.553 | **0.426** | **0.539** | 0.283 | 0.404 | **0.279** | **0.395** | 0.277 | 0.391 | **0.275** | **0.389** |
| | 192 | 0.652 | 0.650 | **0.400** | **0.514** | 0.434 | 0.486 | **0.401** | **0.469** | 0.624 | 0.613 | **0.611** | **0.609** | 0.801 | 0.739 | **0.793** | **0.732** | 0.601 | 0.611 | **0.598** | **0.610** | 0.597 | 0.597 | **0.597** | **0.595** |
| | 336 | 0.629 | 0.640 | **0.386** | **0.502** | 0.447 | 0.506 | **0.392** | **0.487** | 0.647 | 0.641 | **0.645** | **0.642** | 0.863 | 0.766 | **0.842** | **0.757** | 0.633 | 0.637 | **0.630** | **0.637** | 0.635 | 0.625 | **0.633** | **0.624** |
| | 720 | 0.939 | 0.767 | **0.453** | **0.545** | 0.516 | 0.561 | **0.448** | **0.537** | 0.652 | 0.643 | **0.625** | **0.631** | 0.850 | 0.746 | **0.806** | **0.727** | 0.622 | 0.629 | **0.620** | **0.628** | 0.626 | 0.626 | **0.619** | **0.621** |
| | Avg | 0.828 | 0.716 | **0.464** | **0.550** | 0.409 | 0.479 | **0.360** | **0.457** | 0.552 | 0.575 | **0.541** | **0.571** | 0.742 | 0.701 | **0.717** | **0.689** | 0.535 | 0.570 | **0.532** | **0.568** | 0.536 | 0.560 | **0.531** | **0.557** |
| | Improv. (Δ) | - | | 44.0%↑ | 23.2%↑ | - | | 12.0%↑ | 4.6%↑ | - | | 2.0%↑ | 0.7%↑ | - | | 3.4%↑ | 1.7%↑ | - | | 0.6%↑ | 0.4%↑ | - | | 0.9%↑ | 0.5%↑ |

## B.2. Results of Open Benchmarks

Table 14 details the performance across seven real-world datasets characterized by diverse periodicities and data qualities. A key finding is DropoutTS's ability to revitalize legacy architectures: on complex datasets like *Electricity*, it enables the 2021-era **Informer** (Avg MSE **0.489**) to significantly outperform the vanilla Autoformer and rival newer models like TimesNet (0.206). We also observe significant efficacy on the *ILI* dataset, which represents a challenging "small sample, high noise" regime. Here, DropoutTS provides critical regularization, boosting **TimesNet** by **19.4%** and **Crossformer** by **14.2%**, corroborating our claim that spectral sparsity acts as a strong inductive bias aiding generalization when data is scarce. Whether on highly periodic domains (*Weather*) or irregular flows (*ILI*), the consistent positive improvements indicate that the spectral scorer correctly identifies domain-specific "information-rich" frequencies without manual tuning.

## B.3. Detailed Ablation Results

Table 15 offers a granular breakdown of the ablation study, isolating the impact of Detrending strategies and the Spectral Flatness Measure (SFM). The results highlight the critical necessity of Robust Detrending: the "Simple Detrend" (linear interpolation) and "w/o Detrend" variants show severe degradation at high noise levels ($\sigma = 0.9$, MSE degrades to 1.535 vs. 0.990 for Ours), confirming that failing to remove non-stationary trends introduces spectral leakage artifacts that mislead the scorer. Furthermore, the failure of the "w/o SFM Anchor" variant (Avg MSE 1.520) validates that the **SFM** provides an essential physical anchor for distinguishing noise from signal, superior to unconstrained learnable thresholds. Ultimately, our proposed DropoutTS is the only configuration that maintains low error rates monotonically across all noise levels,

*Table 14.* Full forecasting results of real world benchmarks without/with DropoutTS (DT). Better results are in **bold** and underlined. The rows with gray background show the relative improvement (Δ) achieved by our method (higher ↑ is better).

| Method | Metric | Informer (2021) raw MSE | raw MAE | +DT MSE | +DT MAE | Crossformer (2023) raw MSE | raw MAE | +DT MSE | +DT MAE | PatchTST (2023) raw MSE | raw MAE | +DT MSE | +DT MAE | TimesNet (2023) raw MSE | raw MAE | +DT MSE | +DT MAE | iTransformer (2024) raw MSE | raw MAE | +DT MSE | +DT MAE | TimeMixer (2024) raw MSE | raw MAE | +DT MSE | +DT MAE |
|---|---|---|---|---|---|---|---|---|---|---|---|---|---|---|---|---|---|---|---|---|---|---|---|---|---|
| ETTh1 | 96 | 1.569 | 0.902 | 1.305 | 0.833 | 0.394 | 0.404 | 0.384 | 0.397 | 0.394 | 0.392 | 0.385 | 0.390 | 0.488 | 0.475 | 0.456 | 0.454 | 0.384 | 0.391 | 0.382 | 0.390 | 0.401 | 0.395 | 0.392 | 0.389 |
| | 192 | 1.410 | 0.856 | 1.063 | 0.732 | 0.436 | 0.431 | 0.427 | 0.421 | 0.447 | 0.423 | 0.443 | 0.422 | 0.513 | 0.478 | 0.509 | 0.475 | 0.438 | 0.422 | 0.437 | 0.422 | 0.443 | 0.420 | 0.442 | 0.422 |
| | 336 | 1.221 | 0.762 | 0.981 | 0.673 | 0.471 | 0.453 | 0.463 | 0.446 | 0.490 | 0.444 | 0.480 | 0.440 | 0.581 | 0.505 | 0.565 | 0.496 | 0.487 | 0.446 | 0.487 | 0.446 | 0.492 | 0.441 | 0.493 | 0.440 |
| | 720 | 1.147 | 0.773 | 0.960 | 0.734 | 0.495 | 0.493 | 0.490 | 0.490 | 0.506 | 0.470 | 0.477 | 0.464 | 0.556 | 0.509 | 0.549 | 0.506 | 0.481 | 0.466 | 0.475 | 0.466 | 0.496 | 0.460 | 0.493 | 0.459 |
| | Avg | 1.337 | 0.823 | 1.077 | 0.743 | 0.449 | 0.445 | 0.441 | 0.439 | 0.459 | 0.432 | 0.446 | 0.429 | 0.534 | 0.492 | 0.520 | 0.483 | 0.447 | 0.432 | 0.445 | 0.431 | 0.458 | 0.429 | 0.455 | 0.428 |
| | Imp. (Δ) | - | - | 19.5%↑ | 9.7%↑ | - | - | 1.8%↑ | 1.4%↑ | - | - | 2.8%↑ | 0.7%↑ | - | - | 2.6%↑ | 1.8%↑ | - | - | 0.5%↑ | 0.2%↑ | - | - | 0.7%↑ | 0.2%↑ |
| ETTh2 | 96 | 5.257 | 1.710 | 2.659 | 1.243 | 0.339 | 0.395 | 0.306 | 0.354 | 0.293 | 0.338 | 0.288 | 0.333 | 0.448 | 0.428 | 0.405 | 0.409 | 0.298 | 0.340 | 0.296 | 0.338 | 0.297 | 0.338 | 0.293 | 0.337 |
| | 192 | 3.400 | 1.336 | 1.389 | 0.776 | 0.438 | 0.440 | 0.388 | 0.411 | 0.376 | 0.391 | 0.369 | 0.386 | 0.520 | 0.470 | 0.472 | 0.446 | 0.374 | 0.388 | 0.372 | 0.387 | 0.375 | 0.390 | 0.373 | 0.388 |
| | 336 | 1.004 | 0.720 | 0.692 | 0.642 | 0.491 | 0.488 | 0.434 | 0.443 | 0.425 | 0.431 | 0.419 | 0.425 | 0.497 | 0.475 | 0.493 | 0.473 | 0.433 | 0.433 | 0.428 | 0.429 | 0.435 | 0.434 | 0.415 | 0.425 |
| | 720 | 0.969 | 0.713 | 0.830 | 0.648 | 1.913 | 1.072 | 1.378 | 0.878 | 0.441 | 0.450 | 0.437 | 0.448 | 0.456 | 0.465 | 0.455 | 0.461 | 0.432 | 0.443 | 0.428 | 0.441 | 0.437 | 0.447 | 0.437 | 0.447 |
| | Avg | 2.657 | 1.120 | 1.393 | 0.827 | 0.795 | 0.599 | 0.627 | 0.522 | 0.384 | 0.403 | 0.378 | 0.398 | 0.480 | 0.459 | 0.456 | 0.447 | 0.384 | 0.400 | 0.381 | 0.399 | 0.386 | 0.403 | 0.380 | 0.399 |
| | Imp. (Δ) | - | - | 47.6%↑ | 26.2%↑ | - | - | 21.1%↑ | 12.9%↑ | - | - | 1.6%↑ | 1.2%↑ | - | - | 5.0%↑ | 2.6%↑ | - | - | 0.8%↑ | 0.3%↑ | - | - | 1.6%↑ | 1.0%↑ |
| ETTm1 | 96 | 1.502 | 0.870 | 1.224 | 0.788 | 0.345 | 0.362 | 0.329 | 0.355 | 0.327 | 0.348 | 0.326 | 0.351 | 0.455 | 0.439 | 0.446 | 0.434 | 0.330 | 0.351 | 0.329 | 0.350 | 0.318 | 0.342 | 0.316 | 0.341 |
| | 192 | 1.264 | 0.792 | 1.211 | 0.744 | 0.378 | 0.381 | 0.374 | 0.378 | 0.374 | 0.370 | 0.371 | 0.370 | 0.524 | 0.472 | 0.513 | 0.468 | 0.379 | 0.376 | 0.380 | 0.376 | 0.373 | 0.372 | 0.370 | 0.370 |
| | 336 | 1.453 | 0.827 | 1.171 | 0.729 | 0.430 | 0.411 | 0.407 | 0.401 | 0.406 | 0.395 | 0.403 | 0.392 | 0.543 | 0.489 | 0.540 | 0.488 | 0.416 | 0.399 | 0.415 | 0.399 | 0.407 | 0.394 | 0.406 | 0.392 |
| | 720 | 1.700 | 0.929 | 1.170 | 0.780 | 0.498 | 0.463 | 0.464 | 0.434 | 0.477 | 0.433 | 0.471 | 0.430 | 0.554 | 0.495 | 0.557 | 0.497 | 0.473 | 0.433 | 0.473 | 0.434 | 0.474 | 0.432 | 0.473 | 0.430 |
| | Avg | 1.479 | 0.855 | 1.194 | 0.760 | 0.413 | 0.404 | 0.394 | 0.392 | 0.396 | 0.387 | 0.393 | 0.386 | 0.519 | 0.474 | 0.514 | 0.472 | 0.400 | 0.390 | 0.399 | 0.390 | 0.393 | 0.385 | 0.392 | 0.383 |
| | Imp. (Δ) | - | - | 19.3%↑ | 11.1%↑ | - | - | 4.6%↑ | 3.0%↑ | - | - | 0.8%↑ | 0.3%↑ | - | - | 1.0%↑ | 0.4%↑ | - | - | 0.3%↑ | 0.0%↑ | - | - | 0.3%↑ | 0.5%↑ |
| ETTm2 | 96 | 8.137 | 2.167 | 5.555 | 1.739 | 0.181 | 0.264 | 0.178 | 0.264 | 0.175 | 0.252 | 0.175 | 0.252 | 0.207 | 0.282 | 0.206 | 0.282 | 0.180 | 0.257 | 0.180 | 0.256 | 0.173 | 0.250 | 0.173 | 0.250 |
| | 192 | 4.557 | 1.490 | 0.333 | 0.413 | 0.237 | 0.306 | 0.244 | 0.311 | 0.240 | 0.296 | 0.240 | 0.296 | 0.289 | 0.332 | 0.289 | 0.332 | 0.245 | 0.298 | 0.245 | 0.298 | 0.239 | 0.295 | 0.239 | 0.295 |
| | 336 | 0.351 | 0.429 | 0.268 | 0.350 | 0.313 | 0.367 | 0.331 | 0.380 | 0.301 | 0.335 | 0.301 | 0.335 | 0.362 | 0.378 | 0.363 | 0.379 | 0.305 | 0.336 | 0.306 | 0.336 | 0.300 | 0.333 | 0.300 | 0.333 |
| | 720 | 0.384 | 0.425 | 0.381 | 0.451 | 0.784 | 0.642 | 0.650 | 0.556 | 0.402 | 0.394 | 0.402 | 0.394 | 0.534 | 0.463 | 0.534 | 0.464 | 0.406 | 0.394 | 0.406 | 0.394 | 0.398 | 0.391 | 0.398 | 0.391 |
| | Avg | 3.357 | 1.128 | 1.634 | 0.738 | 0.379 | 0.395 | 0.351 | 0.378 | 0.280 | 0.319 | 0.280 | 0.319 | 0.348 | 0.364 | 0.348 | 0.364 | 0.284 | 0.321 | 0.284 | 0.321 | 0.278 | 0.317 | 0.278 | 0.317 |
| | Imp. (Δ) | - | - | 51.3%↑ | 34.6%↑ | - | - | 7.4%↑ | 4.3%↑ | - | - | 0.0%↑ | 0.0%↑ | - | - | 0.0%↑ | 0.0%↑ | - | - | 0.0%↑ | 0.0%↑ | - | - | 0.0%↑ | 0.0%↑ |
| Weather | 96 | 1.094 | 0.753 | 0.763 | 0.596 | 0.166 | 0.204 | 0.162 | 0.200 | 0.173 | 0.208 | 0.165 | 0.202 | 0.181 | 0.227 | 0.184 | 0.230 | 0.185 | 0.217 | 0.187 | 0.219 | 0.180 | 0.226 | 0.164 | 0.202 |
| | 192 | 0.294 | 0.328 | 0.236 | 0.286 | 0.208 | 0.244 | 0.205 | 0.240 | 0.215 | 0.248 | 0.210 | 0.243 | 0.246 | 0.283 | 0.245 | 0.281 | 0.235 | 0.258 | 0.232 | 0.256 | 0.246 | 0.282 | 0.209 | 0.242 |
| | 336 | 0.260 | 0.312 | 0.257 | 0.304 | 0.260 | 0.286 | 0.259 | 0.285 | 0.269 | 0.288 | 0.269 | 0.286 | 0.311 | 0.324 | 0.305 | 0.321 | 0.291 | 0.299 | 0.287 | 0.296 | 0.309 | 0.323 | 0.263 | 0.281 |
| | 720 | 0.289 | 0.336 | 0.267 | 0.305 | 0.331 | 0.339 | 0.330 | 0.339 | 0.345 | 0.337 | 0.343 | 0.334 | 0.417 | 0.387 | 0.403 | 0.378 | 0.362 | 0.345 | 0.362 | 0.345 | 0.395 | 0.375 | 0.339 | 0.332 |
| | Avg | 0.484 | 0.432 | 0.381 | 0.373 | 0.241 | 0.268 | 0.239 | 0.266 | 0.251 | 0.270 | 0.247 | 0.266 | 0.289 | 0.305 | 0.284 | 0.303 | 0.268 | 0.280 | 0.267 | 0.279 | 0.283 | 0.302 | 0.244 | 0.264 |
| | Imp. (Δ) | - | - | 21.3%↑ | 13.7%↑ | - | - | 0.8%↑ | 0.8%↑ | - | - | 1.6%↑ | 1.5%↑ | - | - | 1.7%↑ | 0.7%↑ | - | - | 0.4%↑ | 0.4%↑ | - | - | 13.8%↑ | 12.6%↑ |
| Electricity | 96 | 1.478 | 0.965 | 1.162 | 0.832 | 0.192 | 0.267 | 0.191 | 0.264 | 0.194 | 0.268 | 0.184 | 0.262 | 0.174 | 0.279 | 0.169 | 0.274 | 0.193 | 0.267 | 0.187 | 0.219 | 0.195 | 0.263 | 0.196 | 0.264 |
| | 192 | 1.540 | 0.982 | 0.264 | 0.333 | 0.198 | 0.275 | 0.194 | 0.273 | 0.199 | 0.276 | 0.189 | 0.268 | 0.188 | 0.288 | 0.186 | 0.287 | 0.198 | 0.275 | 0.198 | 0.256 | 0.193 | 0.267 | 0.194 | 0.266 |
| | 336 | 1.412 | 0.951 | 0.263 | 0.332 | 0.214 | 0.295 | 0.211 | 0.290 | 0.214 | 0.292 | 0.213 | 0.291 | 0.210 | 0.307 | 0.208 | 0.305 | 0.214 | 0.292 | 0.214 | 0.293 | 0.209 | 0.283 | 0.209 | 0.283 |
| | 720 | 1.682 | 1.058 | 0.267 | 0.340 | 0.249 | 0.324 | 0.248 | 0.323 | 0.255 | 0.326 | 0.254 | 0.324 | 0.252 | 0.341 | 0.249 | 0.338 | 0.257 | 0.327 | 0.256 | 0.326 | 0.249 | 0.315 | 0.248 | 0.315 |
| | Avg | 1.528 | 0.989 | 0.489 | 0.459 | 0.213 | 0.290 | 0.211 | 0.288 | 0.215 | 0.291 | 0.210 | 0.286 | 0.206 | 0.304 | 0.203 | 0.301 | 0.215 | 0.291 | 0.214 | 0.274 | 0.212 | 0.282 | 0.212 | 0.282 |
| | Imp. (Δ) | - | - | 68.0%↑ | 53.6%↑ | - | - | 0.9%↑ | 0.7%↑ | - | - | 2.3%↑ | 1.7%↑ | - | - | 1.5%↑ | 1.0%↑ | - | - | 0.5%↑ | 5.8%↑ | - | - | 0.0%↑ | 0.0%↑ |
| ILI | 24 | 7.005 | 1.868 | 6.235 | 1.713 | 4.736 | 1.480 | 4.555 | 1.419 | 3.633 | 1.079 | 3.288 | 1.084 | 9.241 | 1.389 | 7.288 | 1.336 | 3.507 | 1.071 | 3.068 | 1.025 | 3.124 | 1.136 | 3.116 | 1.135 |
| | 36 | 7.201 | 1.898 | 5.927 | 1.743 | 5.153 | 1.561 | 4.115 | 1.375 | 4.019 | 1.192 | 3.620 | 1.152 | 7.371 | 1.438 | 5.751 | 1.336 | 3.974 | 1.152 | 3.841 | 1.133 | 3.538 | 1.214 | 3.544 | 1.215 |
| | 48 | 7.173 | 1.920 | 6.348 | 1.784 | 5.244 | 1.576 | 4.031 | 1.359 | 2.939 | 1.099 | 2.874 | 1.093 | 4.175 | 1.237 | 3.272 | 1.120 | 2.513 | 1.005 | 2.468 | 0.991 | 3.055 | 1.130 | 3.064 | 1.129 |
| | 60 | 7.140 | 1.916 | 6.429 | 1.846 | 5.006 | 1.550 | 4.584 | 1.471 | 2.695 | 1.071 | 2.723 | 1.064 | 3.392 | 1.149 | 3.177 | 1.101 | 2.657 | 1.049 | 2.598 | 1.043 | 3.010 | 1.114 | 2.967 | 1.113 |
| | Avg | 7.130 | 1.900 | 6.235 | 1.772 | 5.035 | 1.542 | 4.321 | 1.406 | 3.322 | 1.110 | 3.126 | 1.098 | 6.045 | 1.303 | 4.872 | 1.223 | 3.163 | 1.070 | 2.994 | 1.048 | 3.182 | 1.149 | 3.173 | 1.148 |
| | Imp. (Δ) | - | - | 12.6%↑ | 6.7%↑ | - | - | 14.2%↑ | 8.8%↑ | - | - | 5.9%↑ | 1.1%↑ | - | - | 19.4%↑ | 6.1%↑ | - | - | 5.3%↑ | 2.1%↑ | - | - | 0.3%↑ | 0.1%↑ |

*Table 15.* **Detailed Ablation Study on Synth-12.** Comparison of model performance across noise intensities ($\sigma \in [0.1, 0.9]$). The rightmost column shows the relative improvement of our method compared to the Baseline (arrows indicate performance gain ↑ or drop ↓).

| Method | Configuration Detrend | LogNorm | SFM | $\sigma = 0.1$ MSE | MAE | $\sigma = 0.3$ MSE | MAE | $\sigma = 0.5$ MSE | MAE | $\sigma = 0.7$ MSE | MAE | $\sigma = 0.9$ MSE | MAE | Average MSE | MAE | Δ% (vs. Base) MSE | MAE |
|---|---|---|---|---|---|---|---|---|---|---|---|---|---|---|---|---|---|
| Baseline | - | - | - | 1.228 | 0.862 | 1.189 | 0.846 | 1.180 | 0.843 | 1.140 | 0.831 | 1.060 | 0.798 | 1.159 | 0.836 | - | - |
| Minimal Model | None | ✗ | ✗ | 1.698 | 1.034 | 1.907 | 1.092 | 1.668 | 1.010 | 0.633 | 0.650 | 1.665 | 1.015 | 1.514 | 0.960 | 30%↓ | 14%↓ |
| w/o Detrend+Norm | None | ✗ | ✓ | 2.041 | 1.124 | 1.159 | 0.843 | 1.114 | 0.827 | 1.954 | 1.111 | 1.072 | 0.806 | 1.468 | 0.942 | 26%↓ | 12%↓ |
| w/o Detrend | None | ✓ | ✓ | 1.672 | 1.017 | 1.863 | 1.093 | 1.721 | 1.051 | 1.442 | 0.950 | 1.513 | 0.960 | 1.642 | 1.014 | 41%↓ | 21%↓ |
| Simple Detrend | Simple | ✓ | ✓ | 1.416 | 0.943 | 1.757 | 1.042 | 1.674 | 1.018 | 1.486 | 0.958 | 1.535 | 0.974 | 1.574 | 0.987 | 35%↓ | 18%↓ |
| w/o Spectral Norm | R-OLS | ✗ | ✓ | 1.336 | 0.911 | 1.251 | 0.879 | 1.278 | 0.889 | 1.588 | 0.988 | 1.090 | 0.811 | 1.308 | 0.896 | 12%↓ | 7%↓ |
| w/o SFM Anchor | R-OLS | ✓ | ✗ | 1.386 | 0.933 | 1.532 | 0.968 | 1.540 | 0.977 | 1.693 | 1.029 | 1.448 | 0.944 | 1.520 | 0.970 | 31%↓ | 16%↓ |
| **DropoutTS (Ours)** | R-OLS | ✓ | ✓ | **1.135** | **0.841** | **1.155** | **0.746** | **1.050** | **0.808** | 1.049 | 0.803 | **0.990** | **0.785** | **1.076** | **0.817** | 7.2%↑ | 2.3%↑ |

proving that the synergy between Robust OLS, Log-Norm, and SFM-guided gating is essential for end-to-end robustness.

## B.4. Results on 2025 Architectures and Diverse Domains

Table 16 presents the full per-horizon results on three 2025 state-of-the-art architectures: WPMixer, TimeFilter, and MultiPatchFormer. These models represent distinct design philosophies, including wavelet-based decomposition (WPMixer), Fourier-domain filtering (TimeFilter), and multi-scale patching (MultiPatchFormer). Across ILI and Exchange Rate, DropoutTS yields complementary gains, reducing average MSE by **0.0% to 9.1%**; improvements on ETTh1 are modest (**0.0% to 0.2%**), indicating greater benefits under pronounced distribution shifts. Notably, even architectures with built-in spectral inductive biases (TimeFilter) benefit from our adaptive capacity modulation, suggesting that DropoutTS addresses capacity allocation, a complementary robustness aspect beyond pure architectural design.

*Table 16.* **Full results on 2025 SOTA models.** Per-horizon comparison of three recent 2025 backbones with and without DropoutTS (+DT) on ILI, ETTh1, and Exchange Rate. Better results per model are in **bold**. The gray rows show the relative improvement (Δ) achieved by our method (higher ↑ is better).

| Dataset | H | WPMixer | | | | TimeFilter | | | | MultiPatchFormer | | | |
|---|---|---|---|---|---|---|---|---|---|---|---|---|---|
| | | Raw MSE | Raw MAE | +DT MSE | +DT MAE | Raw MSE | Raw MAE | +DT MSE | +DT MAE | Raw MSE | Raw MAE | +DT MSE | +DT MAE |
| ILI | 24 | 3.173 | 1.022 | 3.018 | 0.987 | 1.903 | 0.859 | 1.720 | 0.821 | 2.933 | 0.938 | 2.110 | 0.897 |
| | 36 | 3.720 | 1.147 | 3.701 | 1.129 | 2.896 | 1.044 | 2.369 | 0.962 | 3.178 | 1.044 | 2.914 | 1.056 |
| | 48 | 2.709 | 1.046 | 2.596 | 1.018 | 2.466 | 0.974 | 2.461 | 0.968 | 2.802 | 1.001 | 2.722 | 1.036 |
| | 60 | 2.722 | 1.044 | 2.591 | 1.017 | 2.361 | 0.983 | 2.342 | 0.975 | 2.303 | 0.982 | 2.452 | 1.015 |
| | Avg | 3.081 | 1.065 | **2.977** | **1.038** | 2.407 | 0.965 | **2.223** | **0.932** | 2.804 | 0.991 | **2.550** | **1.001** |
| | Imp. (Δ) | | - | 3.4%↑ | 2.5%↑ | | - | 7.6%↑ | 3.4%↑ | | - | 9.1%↑ | -1.0%↓ |
| ETTh1 | 96 | 0.388 | 0.386 | 0.386 | 0.385 | 0.390 | 0.390 | 0.387 | 0.388 | 0.391 | 0.383 | 0.395 | 0.385 |
| | 192 | 0.420 | 0.441 | 0.419 | 0.442 | 0.420 | 0.443 | 0.421 | 0.440 | 0.421 | 0.429 | 0.421 | 0.429 |
| | 336 | 0.448 | 0.492 | 0.447 | 0.490 | 0.441 | 0.488 | 0.438 | 0.484 | 0.441 | 0.474 | 0.441 | 0.471 |
| | 720 | 0.470 | 0.501 | 0.471 | 0.500 | 0.462 | 0.486 | 0.461 | 0.485 | 0.463 | 0.477 | 0.460 | 0.469 |
| | Avg | 0.432 | 0.455 | **0.431** | **0.454** | 0.428 | 0.452 | **0.427** | **0.449** | 0.429 | 0.441 | **0.429** | **0.439** |
| | Imp. (Δ) | | - | 0.2%↑ | 0.2%↑ | | - | 0.2%↑ | 0.7%↑ | | - | 0.0%↑ | 0.5%↑ |
| Exchange Rate | 96 | 0.224 | 0.102 | 0.220 | 0.098 | 0.232 | 0.108 | 0.229 | 0.105 | 0.202 | 0.081 | 0.200 | 0.079 |
| | 192 | 0.325 | 0.204 | 0.320 | 0.201 | 0.332 | 0.213 | 0.328 | 0.210 | 0.306 | 0.180 | 0.296 | 0.167 |
| | 336 | 0.454 | 0.393 | 0.453 | 0.391 | 0.456 | 0.396 | 0.455 | 0.394 | 0.414 | 0.321 | 0.411 | 0.316 |
| | 720 | 0.792 | 1.081 | 0.786 | 1.073 | 0.788 | 1.068 | 0.787 | 1.069 | 0.700 | 0.847 | 0.698 | 0.843 |
| | Avg | 0.449 | 0.445 | **0.445** | **0.441** | 0.452 | 0.446 | **0.450** | **0.445** | 0.406 | 0.357 | **0.401** | **0.351** |
| | Imp. (Δ) | | - | 0.9%↑ | 0.9%↑ | | - | 0.4%↑ | 0.2%↑ | | - | 1.2%↑ | 1.7%↑ |

*Table 17.* **Results on Exchange Rate (Finance) and ECG5000 (Healthcare).** MSE comparison of DropoutTS (+DT) with Informer, PatchTST, and TimeMixer on high-noise, non-stationary domains. DropoutTS provides consistent gains even in challenging regimes.

| Dataset | H | Informer | | | PatchTST | | | TimeMixer | | |
|---|---|---|---|---|---|---|---|---|---|---|
| | | Raw MSE | +DT MSE | Imp. (Δ) | Raw MSE | +DT MSE | Imp. (Δ) | Raw MSE | +DT MSE | Imp. (Δ) |
| Exchange Rate | 96 | 0.968 | **0.901** | (↑6.9%) | 0.232 | **0.229** | (↑1.3%) | 0.225 | **0.224** | (↑0.4%) |
| | 192 | 1.033 | **0.916** | (↑11.3%) | 0.335 | **0.331** | (↑1.2%) | 0.325 | **0.324** | (↑0.3%) |
| | 336 | 1.456 | **1.205** | (↑17.2%) | 0.466 | **0.461** | (↑1.1%) | 0.451 | **0.449** | (↑0.4%) |
| | 720 | 1.348 | **1.226** | (↑9.0%) | 0.797 | **0.793** | (↑0.5%) | 0.784 | **0.782** | (↑0.3%) |
| | Avg | 1.201 | **1.062** | (↑11.6%) | 0.458 | **0.454** | (↑0.9%) | 0.446 | **0.445** | (↑0.2%) |
| ECG5000 | 36 | 0.910 | **0.854** | (↑6.1%) | 0.500 | **0.499** | (↑0.2%) | 0.506 | **0.505** | (↑0.2%) |
| | 72 | 0.871 | **0.738** | (↑15.3%) | 0.582 | **0.566** | (↑2.8%) | 0.594 | **0.581** | (↑2.2%) |
| | 144 | 0.824 | **0.812** | (↑1.5%) | 0.630 | **0.615** | (↑2.4%) | 0.638 | **0.633** | (↑0.8%) |
| | 288 | 0.782 | **0.751** | (↑4.0%) | 0.657 | **0.649** | (↑1.2%) | 0.678 | **0.670** | (↑1.2%) |
| | Avg | 0.847 | **0.789** | (↑6.8%) | 0.592 | **0.582** | (↑1.7%) | 0.604 | **0.597** | (↑1.2%) |

To validate generalization beyond standard benchmarks, Table 17 reports results on two challenging domains: Exchange Rate (finance, highly non-stationary) and ECG5000 (Goldberger et al., 2000) (healthcare, irregular cardiac rhythms). On Exchange Rate, DropoutTS reduces Informer MSE by **11.6%**, while gains for PatchTST and TimeMixer remain marginal (**0.9%** and **0.2%**, respectively), highlighting its efficacy on volatile financial series. On ECG5000, PatchTST with DropoutTS achieves a **1.7%** MSE reduction, whereas Informer sees a larger **6.8%** improvement, underscoring its value for medical signals with critical morphology and high noise.

# C. Detailed Visualizations of Signal Regimes

This section visually dissects the "stress test" regimes outlined in the main text. We analyze the spectral behavior of four distinct signal categories under varying noise conditions (Gaussian, Heavy-tail, and Missing Values). These regimes represent a continuum of complexity in real-world time series, ranging from stationarity to evolving spectral dynamics.

## C.1. Stationary Periodic Signals

Figure 8 illustrates the baseline case of stationary signals, which are characterized by time-invariant frequency and amplitude (stable statistical moments). In the frequency domain, this stability manifests as extreme energy concentration, appearing as discrete **Dirac delta peaks** at fundamental frequencies (e.g., 0.5 Hz and 2.0 Hz). This regime typifies systems in stable equilibrium, such as power grid voltage monitoring (fixed daily cycles) or idealized traffic periodicity. Crucially, even under significant noise injection, these sharp spectral peaks remain clearly distinguishable from the broadband noise floor, allowing for near-perfect signal recovery via sparse thresholding.

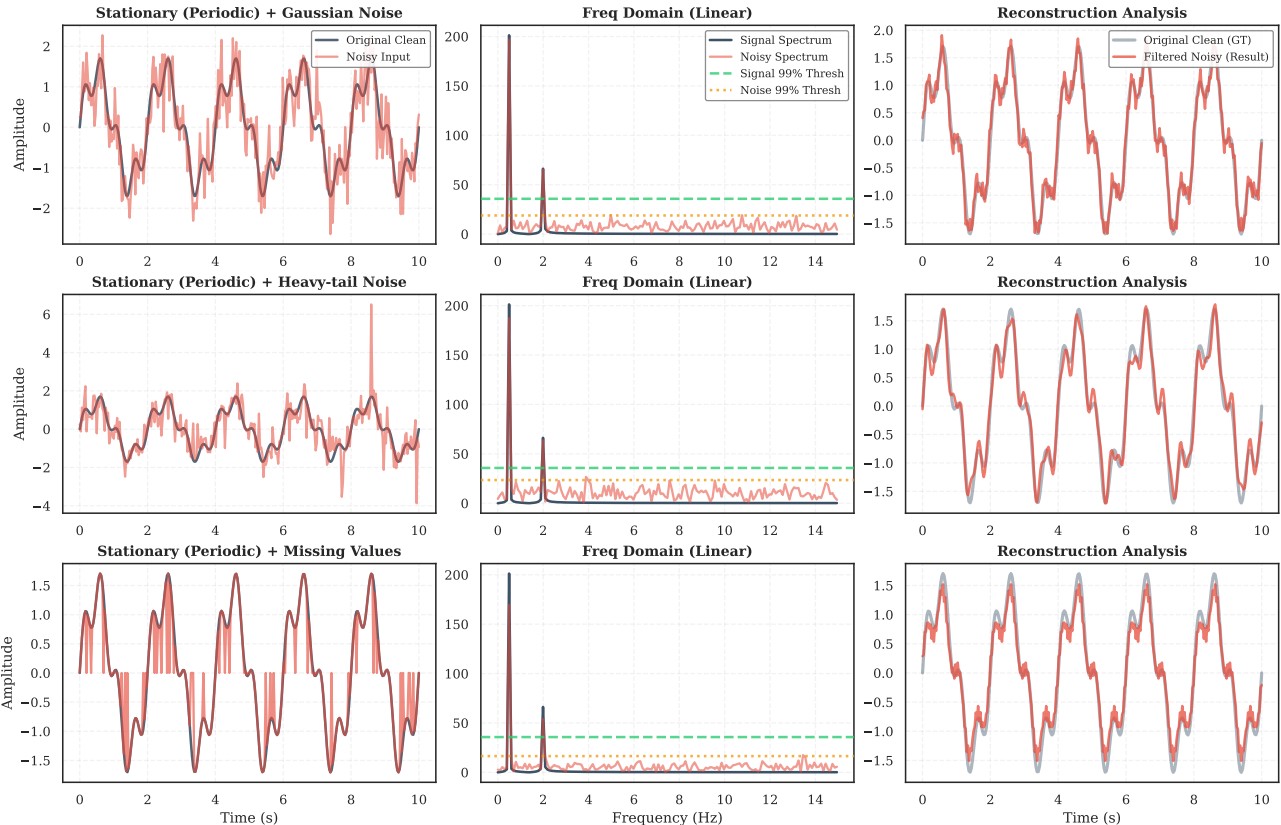

*Figure 8.* **Spectral Analysis of Stationary Periodic Signals.** The frequency spectrum (middle) shows extreme sparsity, with distinct Dirac delta peaks separable from the noise floor (Gaussian/Heavy-tail/Missing). This clarity enables precise thresholding, facilitating faithful ground truth reconstruction (right) despite severe time-domain distortion from high-magnitude artifacts.

### C.2. Non-stationary Mean (Trend & Seasonality)

Figure 9 examines signals with time-varying first moments, combining long-term linear drift with seasonal oscillations. Spectrally, the trend component manifests as **DC Dominance**, a massive concentration of energy at the zero and near-zero frequency components, while seasonality appears as discrete harmonic peaks. This structure is common in macroeconomics (GDP growth), climatology (warming trends), and sales forecasting. Our proposed spectral filtering effectively isolates this low-frequency structure from high-frequency noise, validating the insight that trends constitute a form of "low-frequency sparsity" that can be preserved without complex detrending heuristics.

### C.3. Non-stationary Frequency (Chirp)

Figure 10 presents the challenging spectral drift case (Linear Chirp) with linear frequency sweep (0.5 to 3.0 Hz). Unlike stationary signals, energy forms a **wideband envelope** (structured sparsity) instead of discrete peaks, a pattern critical for industrial vibration monitoring (engine run-up) and biomedical signals. A key observation is that DropoutTS adapts well to this structured sparsity: its thresholding preserves the full energy envelope, maintaining frequency sweep integrity without the phase distortion or amplitude attenuation of fixed low-pass filters.

### C.4. Non-stationary Variance (Amplitude Modulation)

Figure 11 depicts signals with time-varying second moments (Heteroscedasticity), where the amplitude is modulated by a lower-frequency envelope. In the frequency domain, this appears as a carrier peak flanked by distinctive **modulation sidebands**. While time-domain methods often fail here by mistaking low-amplitude segments for noise, the frequency domain representation remains sparse and structured. This allows our method to robustly recover the modulation pattern, a capability essential for financial volatility analysis and mechanical fault diagnosis.

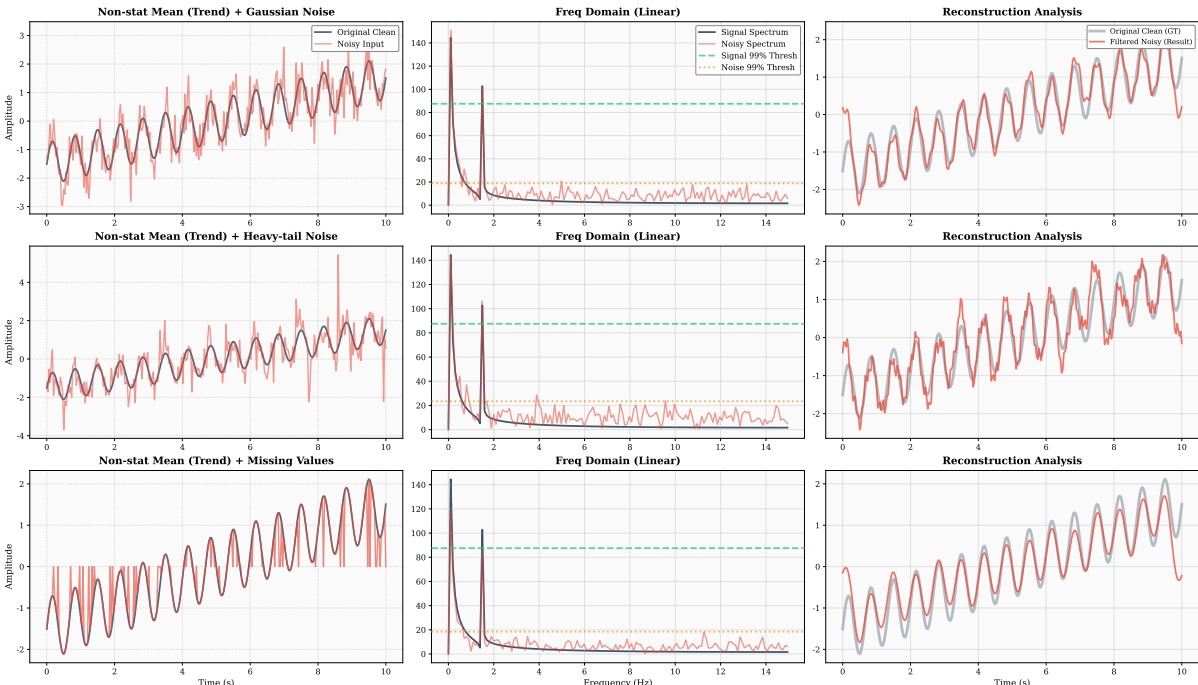

*Figure 9.* **Spectral Analysis of Non-stationary Mean (Trend).** The signal has linear drift and seasonality. The frequency domain shows **DC Dominance** (massive near-zero frequency energy). Our spectral filter (right) preserves low-frequency structure, suppresses wideband noise, and resists trend-noise confusion without manual decomposition.

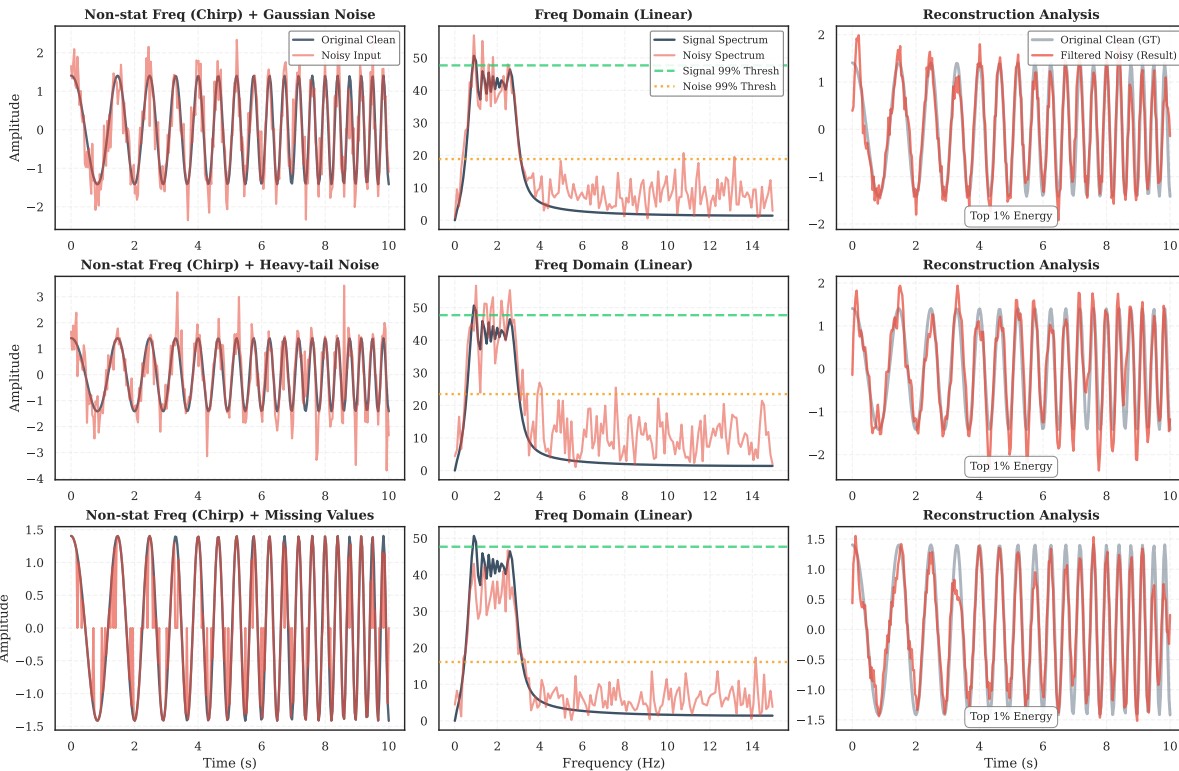

*Figure 10.* **Spectral Analysis of Non-stationary Frequency (Chirp).** The signal sweeps 0.5 to 3.0 Hz, appearing as a **wideband energy block** (structured sparsity) instead of discrete peaks. Unlike phase-distorting static filters, DropoutTS identifies and retains the full spectral envelope (middle), preserving evolving frequency dynamics in the final reconstruction (right) without attenuation.

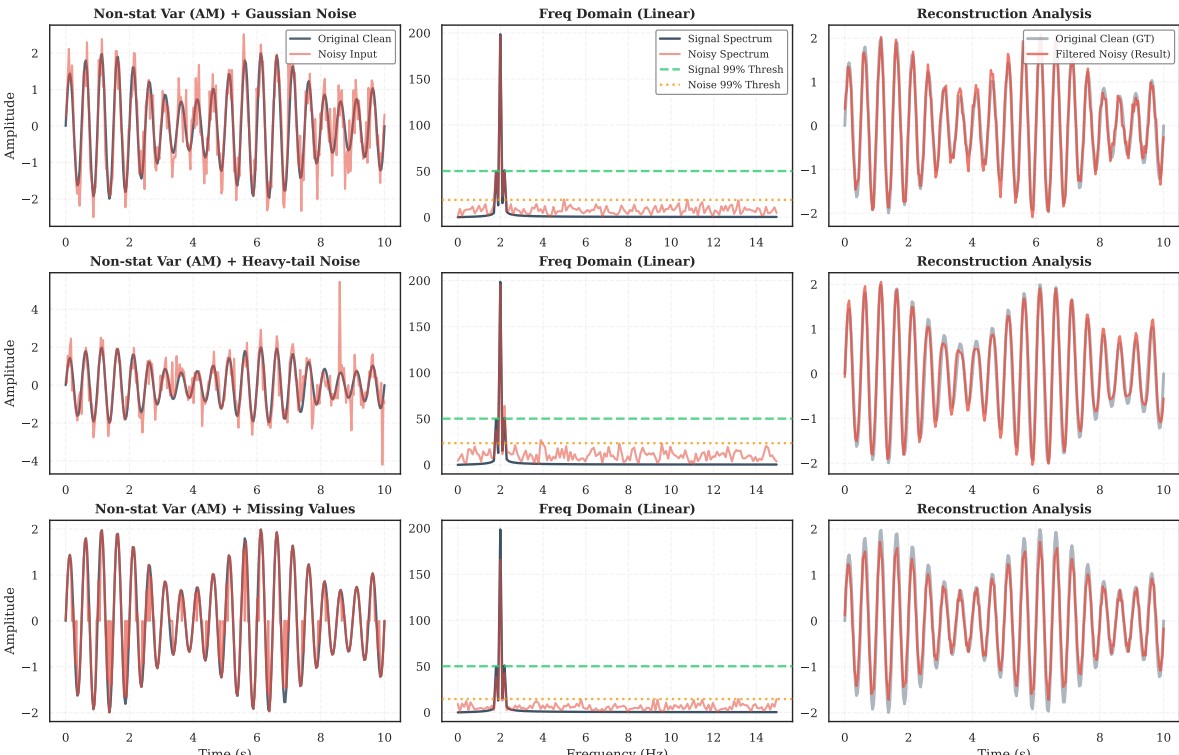

*Figure 11.* **Spectral Analysis of Non-stationary Variance (AM).** The signal features a 2.0 Hz carrier modulated by a 0.2 Hz envelope. The spectrum reveals the characteristic sidebands, which are preserved by the spectral filter to reconstruct the amplitude variations.

## D. Spectral Sparsity Analysis on Real-world Benchmarks

To demonstrate that spectral sparsity is a universal inductive bias rather than an artifact of synthetic data, we conducted a comprehensive spectral analysis on seven representative real-world datasets (ETTh1/h2, ETTm1/m2, Electricity, Weather, ILI). For each dataset, we applied FFT to decompose the series and performed a stress test by retaining only the **top 10%** high-amplitude frequencies (Hard Thresholding) while zeroing out the remaining 90%. We then reconstructed the signal via Inverse FFT. The results, visualized in Figure 12, strongly support the **Sparsity Hypothesis**. The left column displays the time-domain reconstruction, overlaying the original GT (Gray) with the signal recovered from the sparse spectrum (Red). The right column visualizes the underlying log-amplitude spectrum, where the green dashed line explicitly demarcates the boundary between dominant signal components and the discarded noise floor. Across diverse domains, retaining just 10% of the spectral energy yields near-perfect reconstruction (e.g., Correlation $> 0.99$ for *Weather*). This confirms that the removed 90% components largely correspond to high-frequency jitter ("Noise Floor"). Pruning them acts as a natural denoising filter that preserves underlying trends and seasonality, validating the core design of DropoutTS.

### D.1. Empirical Validation of Noise Prevalence and Metric Effectiveness

While the reconstruction visualization confirms the sparsity hypothesis qualitatively, we further conducted a quantitative analysis on the seven benchmark datasets used in our main experiments. This analysis verifies two core premises of DropoutTS: (1) low *Signal-to-Noise Ratios (SNR)* are ubiquitous in real-world time series, motivating the need for robustness; and (2) the *Spectral Flatness Measure (SFM)* serves as a reliable, label-free proxy for ground-truth noise levels.

**Ubiquity of Low SNR.** Since ground-truth noise labels are unavailable in real-world benchmarks, we approximated the SNR based on the verified sparsity hypothesis (treating the top-10% high-energy frequencies as signal and the rest as noise). As shown in Figure 13(a), our analysis reveals a pervasive low-SNR regime. **6 out of 7 datasets** exhibit an estimated SNR below 20 dB (marked by the red dashed line). Specifically, the *Electricity* dataset shows severe noise corruption compared to the cleaner *Weather* dataset. This empirical evidence challenges the implicit clean-data assumption held by many forecasting models and underscores the necessity for robust learning mechanisms.

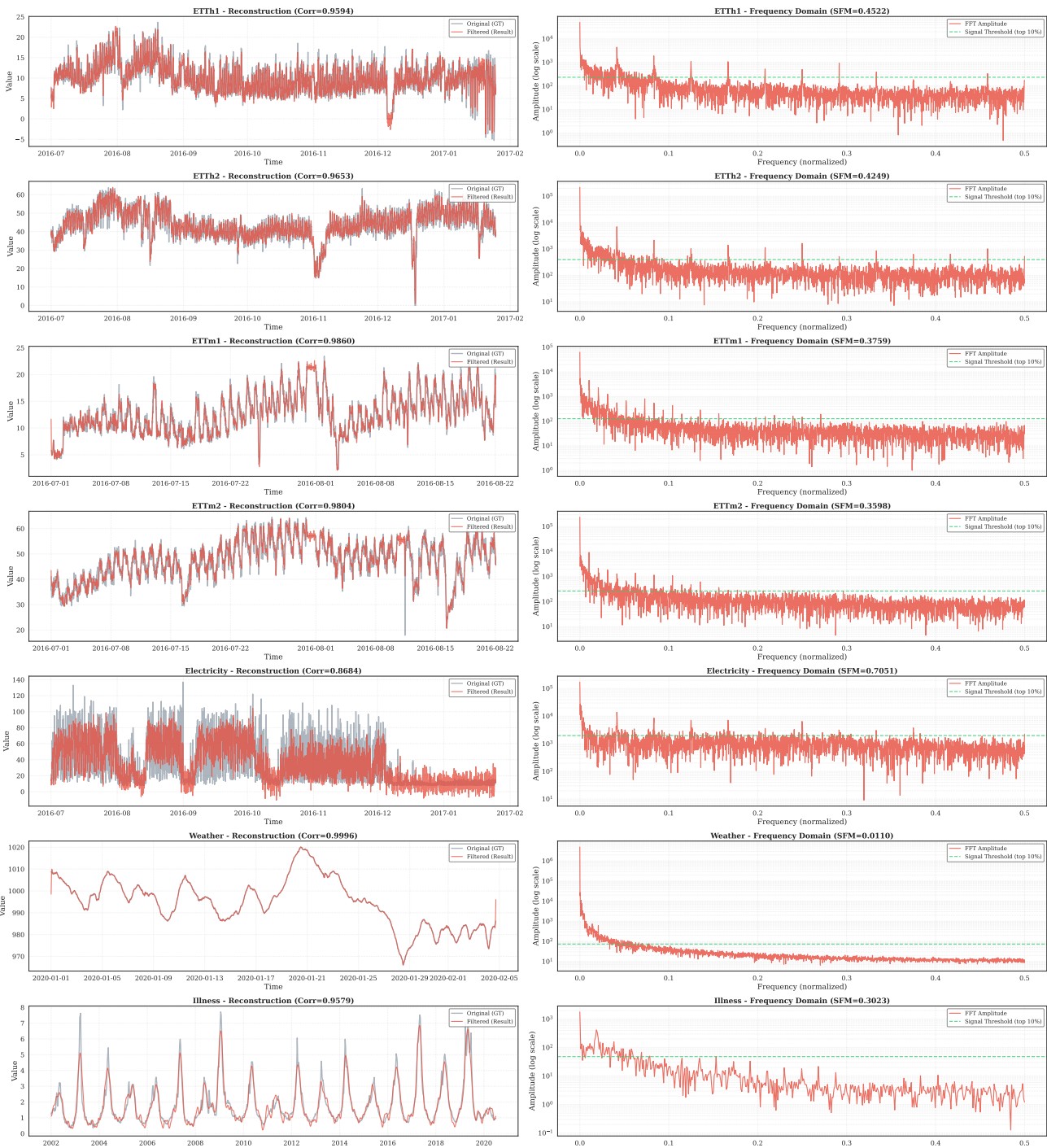

*Figure 12.* **Validation of Spectral Sparsity across 7 Real-world Benchmarks. Left Column:** Time-domain reconstruction comparisons overlaying the original signal (Gray) with the sparse signal recovered from only the top-10% high-energy frequencies (Red). Pearson correlation coefficients (Corr) are annotated in the headers. **Right Column:** Corresponding log-amplitude spectra, where the green dashed line marks the threshold separating the retained dominant signals from the discarded 90% noise floor. The consistently high correlation ($> 0.92$ across all domains) confirms that the majority of valid temporal information is concentrated in a sparse spectral manifold.

**Validity of SFM as a Noise Proxy.** A critical component of DropoutTS is the use of SFM to modulate dropout rates. To validate this design, we analyzed the correlation between the calculated SFM and the estimated SNR across the datasets. Figure 13(b) demonstrates a strong negative linear correlation (Pearson $r = -0.846$, $p < 0.05$). This statistically significant

result confirms that SFM effectively captures the noise intensity: a higher SFM consistently corresponds to a lower SNR.

**Necessity of Adaptive Regularization.** Finally, Figure 13(c) ranks the datasets by their spectral properties, revealing a broad spectrum of noise profiles that range from the clean *Weather* dataset (Low SFM, High SNR) to the noisy *Electricity* dataset. This diversity highlights the "Fixed Dropout Paradox": a static dropout rate cannot simultaneously accommodate such distinct regimes. By mapping SFM to adaptive dropout rates, DropoutTS aligns its regularization strength with the intrinsic difficulty of the data, as evidenced by the inverse relationship between the red (SFM) and green (SNR) bars.

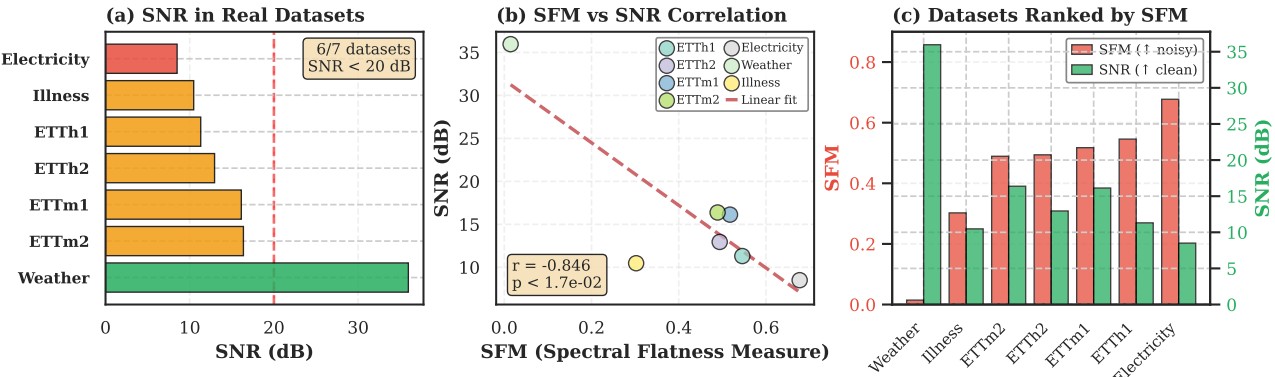

*Figure 13.* **Empirical Motivation and Metric Validation. (a)** Estimated SNR across 7 real-world benchmarks. The prevalence of low-SNR regimes ($< 20$ dB) highlights the ubiquity of noise. **(b)** Correlation analysis between SFM and SNR. The strong negative correlation ($r = -0.846$) validates SFM as a reliable, label-free proxy for noise quantification. **(c)** Dual-view ranking shows significant variance in noise profiles across domains, necessitating the proposed sample-adaptive regularization.

# E. Impact of Spectral Leakage and Detrending

A fundamental prerequisite for frequency-domain analysis via FFT is the *periodicity assumption*, where the signal is treated as one period of an infinite sequence. However, real-world time series are often dominated by non-stationary trends, causing a sharp discontinuity between the last time step $x_L$ and the first $x_1$ (i.e., $|x_L - x_1| \gg 0$).

This discontinuity manifests as **spectral leakage** (Gibbs phenomenon), introducing high-frequency artifacts across the spectrum. If left unaddressed, the spectral mask $\mathbf{M}$ would inadvertently filter out these leakage components, causing the reconstructed signal $\hat{\mathbf{x}}$ to exhibit severe oscillations at the boundaries. This results in an artificially high reconstruction error (specifically **Edge-MAE**) that reflects boundary artifacts rather than true data noise. Thus, **Global Linear Detrending** is a prerequisite to ensure the noise scorer focuses solely on stochastic perturbations.

To validate this necessity, we stress-test three preprocessing strategies: (1) No Detrending, (2) End-to-End Detrending, and (3) Global Linear Detrending across four challenging regimes (Figure 14).

### E.1. Theoretical Comparison

**End-to-End Detrending** is a local approach that estimates the trend solely based on boundary values: $\hat{\mathbf{x}}_{trend}(t) = x_1 + \frac{x_L - x_1}{L} t$. While computationally trivial, it is highly sensitive to sensor noise at the endpoints ($t = 1$ or $t = L$).

**Global Linear Detrending (Ours)** takes a global perspective and, estimates the trend parameters $\mathbf{w}^*, \mathbf{b}^*$ by minimizing the reconstruction error across the entire window as $\mathbf{w}^*, \mathbf{b}^* = \arg\min_{\mathbf{w},\mathbf{b}} \sum_{t=1}^{L} \|\mathbf{x}_t - (\mathbf{w}t + \mathbf{b})\|_2^2$; this formulation ensures the trend estimation captures a *consensus of all data points*, thus yielding strong robustness to local outliers.

### E.2. Analysis of Failure Modes

We evaluate performance via **Edge-MAE**, which quantifies reconstruction fidelity at critical sequence boundaries.

**Sensitivity to Outliers (Row 1).** In the "Linear + Start Outlier" scenario, a single sensor spike at $t = 1$ corrupts the start point. The End-to-End method naively interpolates from this outlier to the end, resulting in a skewed trend line and high error. In contrast, Global Linear Detrending effectively ignores the single outlier, fitting the dominant linear structure.

**Phase Mismatch in Seasonality (Row 2).** Even for periodic signals, if the window length does not match the cycle (Phase Mismatch), $x_1 \neq x_L$. "No Detrending" suffers from severe ringing artifacts. While both detrending methods mitigate this, Global Linear Detrending provides a more stable baseline by minimizing residual energy across the full period.

**Instability under Non-linearity (Row 3).** For quadratic trends, local endpoint noise drastically distorts the end-to-end interpolation slope. In Row 3, minor terminal perturbation causes slope overestimation. Global linear detrending yields the optimal variance-minimizing linear approximation, preserving the intrinsic parabolic residual shape for spectral scoring.

**Robustness to Sensor Failure (Row 4).** This is a critical failure case: a step-up regime shift occurs, but the sensor fails at the final step (dropping to zero). The end-to-end method connects the initial low and failed final low values, falsely inferring a flat trend and fully missing the regime shift. In contrast, global linear detrending leverages the dominant high-state points to accurately estimate an upward trend, demonstrating superior robustness under adversarial conditions.

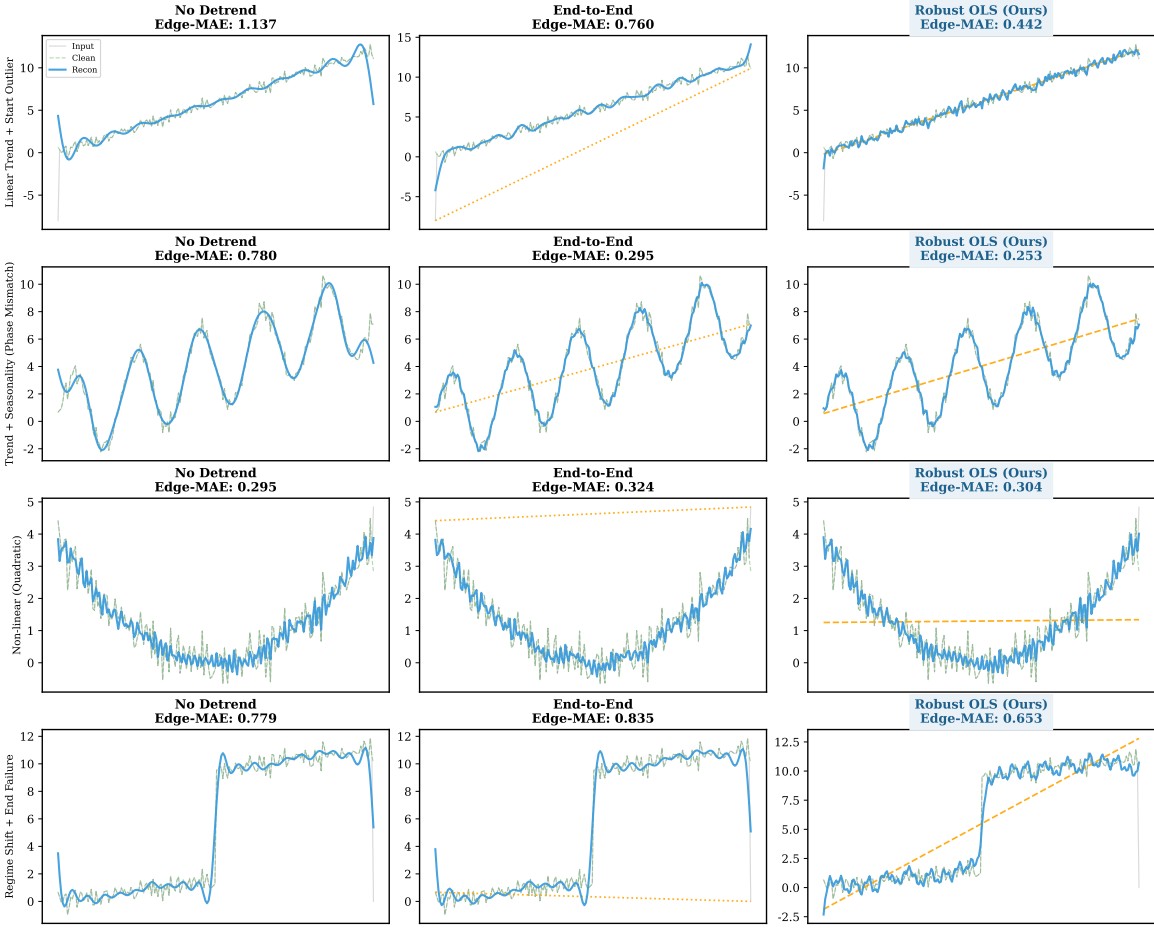

*Figure 14.* **Impact of Detrending Strategies.** We analyze signal reconstruction quality (Edge-MAE) under four stress tests. Red circles highlight failures. **Row 1 (Outlier):** End-to-End is biased by a start-point outlier; Global OLS remains stable. **Row 2 (Seasonality):** Detrending is essential to prevent spectral leakage caused by phase mismatch. **Row 3 (Non-linear):** Endpoint noise causes End-to-End to estimate an incorrect slope. **Row 4 (Regime Shift + Failure):** *Critical Failure Case.* A sensor failure at the last step causes End-to-End to miss the upward trend entirely (inferring a flat line). Global Linear Detrending correctly captures the global upward shift despite the endpoint failure. **Conclusion:** DropoutTS (Global Linear Detrending) consistently achieves the lowest boundary error.

# F. Theoretical Analysis

## F.1. Formalization of the Fixed Dropout Paradox

To theoretically ground the limitation of static regularization, we analyze a simplified linear regression setting with heteroscedastic noise. This proxy enables tractable analysis of the inherent capacity mismatch in deep forecasting models.

**Setup.** Consider a linear model $y = \mathbf{w}^{*\top}\mathbf{x} + \epsilon$, where input $\mathbf{x} \in \mathbb{R}^d$ and noise $\epsilon \sim \mathcal{N}(0, \sigma^2(\mathbf{x}))$. The noise intensity $\sigma(\mathbf{x})$ is input-dependent (heterogeneous). Following Wager et al. (2013), training with Dropout rate $p$ effectively imposes an adaptive $L_2$ regularization. Under the assumption of isotropic features, the objective approximates:

$$\mathcal{L}(\mathbf{w}; p) \approx \frac{1}{n}\|\mathbf{y} - \mathbf{X}\mathbf{w}\|_2^2 + \lambda(p)\|\mathbf{w}\|_2^2 \tag{12}$$

where $\lambda(p) = \frac{p}{1-p}$ controls the regularization strength.

The generalization excess risk can be characterized by the bias-variance decomposition. In the small regularization regime, the risk scales as (Hastie et al., 2009):

$$\mathcal{E}(p, \sigma) \sim \underbrace{\lambda(p)^2 C_1}_{\text{Bias}^2} + \underbrace{\frac{\sigma^2}{(1 + \lambda(p))^2} C_2}_{\text{Variance}} \tag{13}$$

where $C_1, C_2$ are data-dependent constants. *Note: A higher dropout rate $p$ increases $\lambda(p)$, which suppresses Variance (noise fitting) at the cost of increased Bias.*

**Theorem F.1** (Sub-optimality of Fixed Dropout). *Assume the risk function $\mathcal{E}(\lambda, \sigma)$ is strictly convex with respect to $\lambda$. The optimal regularization strength $\lambda^*(\sigma)$ scales with the noise level $\sigma$ (i.e., $\lambda^*$ is monotonic increasing w.r.t $\sigma$). Consequently, the sample-optimal dropout rate $p^*(\sigma)$ is strictly input-dependent.*

*For a dataset with heterogeneous noise distributions (i.e., $Var[\sigma(\mathbf{x})] > 0$), applying a single fixed dropout rate $p_{fix}$ incurs a strictly positive sub-optimality gap compared to the optimal adaptive strategy:*

$$\Delta\mathcal{R} = \mathbb{E}_{\mathbf{x}}\left[\mathcal{E}(p_{fix}, \sigma(\mathbf{x})) - \mathcal{E}(p^*(\sigma(\mathbf{x})), \sigma(\mathbf{x}))\right] > 0 \tag{14}$$

*Proof Sketch.* Since $\mathcal{E}$ is convex w.r.t $\lambda$ and the optimal $\lambda^*$ varies with $\mathbf{x}$, by Jensen's inequality, a single fixed parameter cannot simultaneously minimize the risk for samples with distinct noise levels. $\square$

**Remark.** DropoutTS approximates this optimality by dynamically aligning $p(\mathbf{x})$ with the spectral noise score $s(\mathbf{x}) \propto \sigma(\mathbf{x})$, theoretically minimizing the point-wise risk.

### F.2. Generalization under Spectral Sparsity

We extend analysis to time series models via Rademacher complexity, linking spectral properties to generalization bounds.

**Assumption (Spectral Sparsity).** The clean signal $\mathbf{x}^\star$ can be approximated by a subspace $\mathcal{M}_K$ spanned by $K$ dominant frequencies, where $K \ll L$ (sequence length).

**Theorem F.2** (Adaptive Generalization Bound). *Let $\mathcal{H}$ be a hypothesis class of $\mu$-Lipschitz forecasters. Under input-dependent dropout $p(\mathbf{x})$, the generalization error $\mathcal{R}(f)$ is bounded with probability at least $1 - \delta$ by:*

$$\mathcal{R}(f) \leq \hat{\mathcal{R}}(f) + \frac{2\mu}{\sqrt{n}}\sqrt{\frac{1}{n}\sum_{i=1}^{n}(1 - p(\mathbf{x}_i))^2 K} + 3\sqrt{\frac{\log(2/\delta)}{2n}} \tag{15}$$

*where $\hat{\mathcal{R}}(f)$ is the empirical risk and the second term represents the effective Rademacher complexity controlled by dropout.*

**Implication.** The complexity term depends on the effective capacity $\sqrt{\frac{1}{n}\sum(1 - p(\mathbf{x}_i))^2 K}$.

- For **noisy samples** (corrupted spectral structure), DropoutTS assigns $p(\mathbf{x}_i) \to p_{\max}$, effectively removing these samples from the complexity budget to prevent overfitting broadband noise.

- For **clean samples**, $p(\mathbf{x}_i) \to p_{\min}$, allowing the model to utilize full capacity ($K$ frequencies) to capture the signal.

This adaptive mechanism yields a strictly tighter bound than uniform regularization, which would either under-regularize noisy samples or over-penalize clean signals.

