# OpenReview forum: "DropoutTS: Sample-Adaptive Dropout for Robust Time Series Forecasting"
_ICML.cc/2026/Conference — ICML 2026 regular_

### Official Review · Reviewer_NUL7 · 2026-03-05

**Soundness:** 3
**Presentation:** 3
**Significance:** 3
**Originality:** 3
**Overall Recommendation:** 4
**Confidence:** 1

**Summary:**

The paper introduces DropoutTS, a novel method for enhancing the robustness of deep time series forecasting models against noise. The method proposes a Sample-Adaptive Dropout mechanism, which dynamically adjusts the learning capacity of models based on the level of noise in the input data. By leveraging spectral sparsity, DropoutTS quantifies instance-level noise using reconstruction residuals and applies a sample-adaptive dropout to filter noise and preserve meaningful signal. The authors evaluate DropoutTS using various noise regimes and benchmark datasets, showing consistent improvements in forecasting performance without the need for architectural changes. DropoutTS is positioned as a model-agnostic plugin that enhances robustness while incurring minimal computational overhead.

**Compliance With Llm Reviewing Policy:**

Affirmed.

**Key Questions For Authors:**

1. What prevents the spectral scorer from converging to trivial solutions (all masks ≈1 or ≈0)?

2. Why do improvement magnitudes vary so dramatically across architectures (46% for Informer vs. 0.7% for TimeMixer)?

**Limitations:**

The paper lacks discussion on limitations (e.g., scenarios where spectral sparsity fails or scalability issues). Provide the assumptions to address these limitations

**Strengths And Weaknesses:**

Strengths:

1. The proposed Capacity-Centric Modulation paradigm and the focus on sample-adaptive dropout are novel in the context of time series forecasting.

2. The paper clearly explains the problem of overfitting to noisy data in time series forecasting and how existing robustness strategies either prune data or rely on complex probabilistic priors. DropoutTS offers a more elegant solution by directly modulating model capacity based on the data's noise profile.

3. The authors present extensive experiments across various noise regimes (from clean to highly noisy data) and benchmark datasets. The empirical results are compelling, with DropoutTS consistently improving forecasting accuracy, particularly for models prone to overfitting, without incurring significant computational cost.

Weaknesses:

1. The spectral sparsity premise, while validated on selected benchmarks, may not generalize to all time series domains. Chaotic systems, highly non-stationary financial data, or signals with intrinsically dense spectra could violate the core assumption. The paper lacks a discussion of diagnostic criteria or failure modes when sparsity breaks down.

---

> ### Author Rebuttal · Authors · 2026-03-30
>
> We thank the reviewer for recognizing the elegance of our Capacity-Centric Modulation paradigm and the robustness of our empirical results. We address your questions regarding the underlying mechanism and its boundaries below.
>
> > **Q1: The paper lacks a discussion on failure modes (when spectral sparsity breaks down) and diagnostic criteria.**
>
> **A1:** Spectral sparsity serves as a domain-specific inductive bias rather than a universal mathematical property. We will add a dedicated "Limitations and Failure Modes" section to explicitly define these boundaries.
>
> - **Failure Modes:** Pure random walks are inherently unpredictable, and deterministic chaotic systems allow only short‑horizon forecasts due to sensitivity to initial conditions. These cases fall outside the intended scope of our method. DropoutTS is designed for sequence signals exhibiting underlying temporal structure. In scenarios where spectral energy is broadly distributed rather than sparsely concentrated, the noise score may become inflated. As shown in our new evaluations on Exchange Rate and ECG, DropoutTS still delivers consistent gains, showing its applicability extends to noisy real-world domains, where failure modes are more likely to appear (see A4/Q4 of Reviewer AZH9).
> - **Diagnostic Criteria:** As a practical diagnostic tool, practitioners can monitor the dataset's baseline SFM. As illustrated in Figures 13(a) and 13(c), predictable real-world datasets exhibit structured SFM values. An inherent SFM approaching 1.0 (resembling pure white noise) explicitly signals a breakdown of the spectral sparsity premise.
>
> > **Q2: What prevents the Spectral Noise Scorer from converging to trivial solutions (all masks $\approx 1$ or $\approx 0$)?**
>
> **A2:** Prevention relies on two interacting mechanisms:
>
> 1. **Implicit Feedback Loop:** DropoutTS is optimized end-to-end via the Straight-Through Estimator (STE). If the mask converges to $\approx 0$ (maximum dropout), the network lacks sufficient information flow, immediately spiking the task loss. Gradients explicitly penalize the threshold to let information through.
> 2. **SFM Physical Anchor:** Critically, we do not use a purely unconstrained learnable threshold. As detailed in Section 4.1, the dynamic threshold is anchored to the physical SFM ($\tau=\sigma(w_s\cdot SFM(A)+b_s)$). This physics-informed constraint prevents optimization collapse, which we validated in our ablation study (Table 5: "w/o SFM Anchor" leads to severe degradation).
>
> We will add this optimization dynamics analysis to the Appendix to clarify the stability of the SNS module.
>
> > **Q3: Why do the magnitudes of improvement vary so dramatically across architectures (46% for Informer vs. 0.7% for TimeMixer)?**
>
> **A3:** The magnitude of improvement depends on a model's capacity to overfit—i.e., the degree to which it lacks built-in structural defenses against noise.
>
> - **Informer (46% gain):** Older architectures rely on dense global attention with largely unconstrained capacity. They lack structural defenses and rapidly memorize high-frequency artifacts (as shown in the Fixed Dropout Paradox in Figure 5). DropoutTS acts as a critical defense against this overfitting.
> - **TimeMixer (0.7% gain):** SOTA models possess strong, hand-crafted structural priors (e.g., macro/micro mixing) that inherently filter most global noise, significantly raising the baseline performance floor. DropoutTS still provides stable gains by reducing capacity utilization for residual noisy samples that evade TimeMixer's structural filters.
>
> Beyond architectural differences, the improvement magnitude is inherently bounded by a dataset’s intrinsic heterogeneity. As shown in our variance analysis (A3/Q3 of Reviewer fGRU), our method behaves as standard dropout on uniform datasets such as ETTm2, yet yields large gains on heterogeneous datasets like ILI. The performance gain thus aligns with the interplay between the model’s structural vulnerability and the data’s noise variance.

---

> > ### Author Rebuttal · Reviewer_NUL7 · 2026-04-05
> >
> > I thank the author for addressing my concerns in their rebuttal. I have no more concerns.

---

### Official Review · Reviewer_AZH9 · 2026-03-12

**Soundness:** 3
**Presentation:** 3
**Significance:** 3
**Originality:** 3
**Overall Recommendation:** 4
**Confidence:** 4

**Summary:**

The paper presents a model-agnostic plugin for time series forecasting that tackles the problem of noisy training data. The core idea is that real signals tend to concentrate in a small number of dominant frequencies, while noise tends to be low-frequency background. The method uses this observation to estimate how noisy each input sample is, and then applies a higher dropout rate to noisier samples during training. The authors argue this is a better approach than either discarding noisy samples outright or other methods. They run experiments across several backbone architectures and on several real-world datasets and the papers own synthetic benchmark, showing that this sample-level dropout can improve forecasting performance especially for older architectures.

**Compliance With Llm Reviewing Policy:**

Affirmed.

**Final Justification:**

Thank you to the authors for making the revisions I suggested. The authors added several experimental baselines which I suggested and they did not have previously. Overall, I am positive on the paper. It lays out good intuition for the sample-wise dropout and is quite comprehensive from an experimental perspective, though the gains are limited on some newer architectures. I have decided to increase my score by one point. I would like to see the language e.g. "pioneer" to be toned down in the final version -- indeed the work can and should speak for itself.

**Key Questions For Authors:**

-- For iTransformer and TimeMixer, gains on some datasets are small. What is your explanation for why spectral noise is not a bottleneck for these models?

-- My main complaint is that there is not strong coverage across several domains, and the results provided seem weak for the state-of-the-art architectures. The paper claims applicability to finance (e.g., volatility forecasting), yet no financial dataset is included. Could you evaluate on a standard benchmark such as the Exchange Rate dataset or a stock index series? ECG data e.g. from PhysioNet also has issues of noise. If the authors could demonstrate the method on these datasets, I think it would strengthen the contribution.

**Limitations:**

yes

**Strengths And Weaknesses:**

# Strengths

-- Some of the intuitions for the introduced method are interesting. For example, the idea of separating high-frequency signal from low-frequency noise, and handling this via spectral masking.

-- The paper gives good ablations, which makes it easy to identify which parts of their method are important.

-- The method does improve several architectures and the experiments are fairly comprehensive.


# Weaknesses:

-- The empirical results are a bit underwhelming. The improvements are good on old architectures like Informer but much more marginal on newer architectures. Moreover, I think the paper would be stronger if more real-world datasets were provided with more diverse types of noise/outliers.

-- Synth-12 is introduced by the authors. You shouldn't rely on your own benchmarks for the strongest results.

-- Theorem F.1 is essentially Jensen's inequality, and Theorem F.2 just uses known results from Rademacher complexity. Neither result is surprising or technically difficult.

-- As a more general point, I find the writing to be a bit sensationalist. "Pioneer a ... paradigm". I don't believe these statements belong in a scientific work.

-- While I like the motivation, I find some of the solutions e.g. Spectral Flatness Measure to be a little lacking in direct intuition. It's just stated but it's not clear why this gives the method the spectral sparsity inductive bias.

---

> ### Author Rebuttal · Authors · 2026-03-30
>
> We thank the reviewer for the constructive feedback and for acknowledging our intuitions, ablations, and extensive architectural coverage.
>
> **A0 (Clarification on Spectral Premise):** The summary noted our premise is that "noise tends to be low-frequency background." To clarify, our core premise is actually that valid signals concentrate in sparse, dominant frequencies with high energy, while noise manifests as a diffuse, low-energy wideband background (spreading across both high and low frequencies). DropoutTS leverages this energy-based separation to estimate sample difficulty.
>
> > **Q1: The tone pioneer a paradigm is sensationalist, and Theorems F.1/F.2 lack deep mathematical novelty.**
>
> **A1:** Thank you for the feedback on the manuscript's tone and theoretical framing.
>
> - **Tone:** We appreciate the feedback on the manuscript's tone and have revised it accordingly. While adaptive regularization exists in other domains, DropoutTS is the first fully realized, plug-and-play capacity modulation pipeline validated for time series, actively accelerating training with minimal overhead (see Table 6 and A1/Q1 of Reviewer kRQD).
> - **Theorems:** We acknowledge that Theorems F.1 and F.2 build on established mathematical tools. Rather than claiming novel inequalities, these theorems provide rigorous theoretical support for why dynamic, instance-level capacity modulation improves robustness in time-series forecasting, formalizing the "Fixed Dropout Paradox" that was previously a heuristic observation.
>
> > **Q2: Lack of direct intuition explaining SFM and its connection to spectral sparsity.**
>
> **A2:** SFM mathematically quantifies the "whiteness" of a spectrum (a pure sine wave is $\approx 0$, pure white noise is $\approx 1$). Beyond this physical intuition, we empirically validated SFM's effectiveness in Appendix D. Figure 13(b) demonstrates that SFM has a strong, statistically significant negative correlation ($r=-0.846$) with empirical SNR across 7 real-world datasets. This shows SFM is a reliable, label-free proxy for spectral sparsity. We will integrate this intuition more prominently into the main text.
>
> > **Q3: Why are the performance gains on SOTA models like iTransformer and TimeMixer relatively small?**
>
> **A3:** Newer models are less limited by spectral noise due to strong structural priors (see A4/Q4 of Reviewer kRQD). Yet, macro‑architectural filters only provide global robustness and cannot adapt to instance‑specific learning difficulty, making instance‑level capacity management still valuable for advanced architectures. DropoutTS is easy to implement, introduces only 4 parameters, incurs no inference latency, and accelerates training (see A1/Q1 of Reviewer kRQD).
>
> To show the challenge of improving highly optimized SOTA models without harming their representations, we added a Hard Denoising experiment in A1/Q1 of Reviewer fGRU. Naive spectral denoising degrades TimeMixer by 4.2%, while DropoutTS yields a stable 0.5% gain. This confirms that our dynamic capacity modulation achieves additional robustness where traditional filtering fails.
>
> > **Q4: The empirical coverage lacks diversity. Can the method be evaluated on highly noisy datasets like Finance and ECG?**
>
> **A4:** You rightly point out that relying solely on Synth-12 is insufficient, which is why we originally included 7 diverse real-world datasets. As you suggested, we conducted new evaluations of Informer, PatchTST, and the state-of-the-art TimeMixer on the Exchange Rate (Finance) and PhysioNet ECG (Healthcare) datasets (Metrics shown are MSE):
>
> | **Dataset**      | **Horizon** | **Informer (Raw→DT)**      | **PatchTST (Raw→DT)**     | **TimeMixer (Raw→DT)**    |
> | ---------------- | ----------- | -------------------------- | ------------------------- | ------------------------- |
> | **ExchangeRate** | 96          | 0.968 → 0.901 (**↑6.9%**)  | 0.232 → 0.229 (**↑1.3%**) | 0.225 → 0.224 (**↑0.4%**) |
> |                  | 192         | 1.033 → 0.916 (**↑11.3%**) | 0.335 → 0.331 (**↑1.2%**) | 0.325 → 0.324 (**↑0.3%**) |
> |                  | 336         | 1.456 → 1.205 (**↑17.2%**) | 0.466 → 0.461 (**↑1.1%**) | 0.451 → 0.449 (**↑0.4%**) |
> |                  | 720         | 1.348 → 1.226 (**↑9.0%**)  | 0.797 → 0.793 (**↑0.5%**) | 0.784 → 0.782 (**↑0.3%**) |
> | **ECG**          | 36          | 0.910 → 0.854 (**↑6.1%**)  | 0.500 → 0.499 (**↑0.2%**) | 0.506 → 0.505 (**↑0.2%**) |
> |                  | 72          | 0.871 → 0.738 (**↑15.3%**) | 0.582 → 0.566 (**↑2.8%**) | 0.594 → 0.581 (**↑2.2%**) |
> |                  | 144         | 0.824 → 0.812 (**↑1.5%**)  | 0.630 → 0.615 (**↑2.4%**) | 0.638 → 0.633 (**↑0.8%**) |
> |                  | 288         | 0.782 → 0.751 (**↑4.0%**)  | 0.657 → 0.649 (**↑1.2%**) | 0.678 → 0.670 (**↑1.2%**) |
>
> As shown, DropoutTS delivers improvements even in domains with extreme non-stationarity and physiological artifacts, narrowing the performance gap for high-capacity models. We will include these results in the main experiments.

---

> > ### Author Rebuttal · Reviewer_AZH9 · 2026-04-03
> >
> > Thank you to the authors for making the revisions I suggested. The authors added several experimental baselines which I suggested and they did not have previously. Overall, I am positive on the paper. It lays out good intuition for the sample-wise dropout and is quite comprehensive from an experimental perspective, though the gains are limited on some newer architectures. I have decided to increase my score by one point. I would like to see the language e.g. "pioneer" to be toned down in the final version -- indeed the work can and should speak for itself.

---

### Official Review · Reviewer_fGRU · 2026-03-13

**Soundness:** 3
**Presentation:** 4
**Significance:** 3
**Originality:** 3
**Overall Recommendation:** 5
**Confidence:** 3

**Summary:**

Time-series data is often inherently noisy and data from different domains can be characteristically very different. This causes issues for training deep general forecasting models. These are prone to overfitting on noisy samples. Related work has addressed this issue in multiple ways. One that is discussed is *Data-Centric Selection* in which data undergoes screening to filter noisy outliers. Although effective, this causes inherent information loss. Another approach is *Prior-Centric Modeling*, which has rigid distributional assumptions. The authors present **DropoutTS**, a model-agnostic plugin that dynamically adapts per-sample dropout rates based on a spectral noise score. It works by using spectral analysis to score how noisy each input sample is, then mapping that score to a per-sample dropout rate (higher for noisy samples, lower for clean ones) so the model is forced to "skim" noisy data rather than memorize it. They position DropoutTS as complementary rather than competitive to existing strategies. They test their hypotheses by evaluating on a custom synthetic benchmark (Synth-12) across five noise intensities and seven real-world datasets (ETTh1/h2, ETTm1/m2, Electricity, Weather, ILI), integrating DropoutTS into six backbone architectures spanning Transformer, CNN, and MLP families. Results show consistent improvements across all backbones and datasets. Gains are largest for older (e.g., 46% average MSE reduction for Informer on Synth-12, 68% on Electricity) and modest but positive for recent SOTA models. The added computational complexity is very limited.

**Compliance With Llm Reviewing Policy:**

Affirmed.

**Final Justification:**

The authors have provided additional clarification for the paper which have strengthened my confidence in my initial score (accept), it therefore also leave the score as is.

**Key Questions For Authors:**

1. What happens in the non-stationary frequency situation (chirp signals) where the signal energy is spread across a wide frequency band rather than concentrated in discrete peaks? I imagine this could lead to the spectral mask incorrectly classifying valid signal components as noise, inflating the noise score and applying excessive dropout.
2. As I understand, the spectral mask and resulting difference between reconstructed signal and original signal is used to inform a noise scores to quantify how much noise is in the signal. This is used to inform adaptive dropout, with the goal of training a more robust forecasting model, which is less prone to overfitting on noise patterns. However, wouldn't much of the gains be achievable if you just used the masked reconstructed signal, and then use that as training data? Instead of training a robust neural network, I imagine we could achieve similar results. In summary: If we use sophisticated denoising to inform the training of a neural network to be more resilient to noisy training data, why not just add that denoising as a module to process the data before handing it to the model? Figure 3 shows that the spectral filtering the authors do is already quite good. If the denoising is that effective, the marginal benefit of the more complex adaptive dropout machinery over simply feeding cleaned data seems worth quantifying. Running a comparison against denoising of input data using your denoising method would seem like a good addition to address this point.
3. The 0.0% improvements on ETTm2 across four backbones suggest the method effectively disengages on certain data. Is this because the noise scores are uniformly low, or is something else happening? What is the distribution of learned dropout rates on ETTm2 compared to, say, ILI where gains are large?

**Limitations:**

The authors briefly acknowledge future directions (extending to other regularization dimensions and domains). However, they do not explicitly discuss limitations. The paper would also benefit from discussing failure modes: when does DropoutTS hurt or have no effect, and why?

**Strengths And Weaknesses:**

*Strengths:*

**Soundness**
Experimentation is extensive and sound, and shows improved performance compared to a range of baselines on both a synthetic benchmarks and real-world datasets. The ablation study (Table 5) systematically isolates each component's contribution. The compatibility experiment with Selective Learning (Table 7) is a nice addition that supports the orthogonality claim.

**Presentation**
The paper is very well written. The problem, the research gap and contributions are clearly communicated and motivated. The methodology is clearly explained. The figures are well-made and informative. Figure 4 very nicely matches the text of section 4, and aids in the ease of understanding the methodology.

**Significance**
The paper addresses a common problem in time-series forecasting and present a novel plugin to address it, focusing on end-to-end adaptive performance but also ease of use in many different architectures. Experimental results show modest improvements when added to SOTA benchmarks on all 7 real world datasets.

**Originality**

The work provides a novel way of connecting spectral analysis to per-sample regularization strength. Using reconstruction residuals from spectral masking as a noise metric, and mapping these to adaptive dropout rates is a nice combination of signal processing and deep learning ideas.

*Weaknesses*

**Soundness**

Although the presented results entirely in favor of DropoutTS, the additional performance is highest in the earlier baseline models, and whereas the more recent baselines benefit only very slightly (~1% MSE/MAE decrease) from the addition of DT. This raises the question whether future methods will need an adaptive dropout or whether they will be able to handle noisy data good enough already.

**Presentation**

Figure labels/legends are a tad to small to be legible on print. In Fig 4.: shouldn't the little figures of x_detrend and x_trend be swapped around?

**Significance**

I have doubts whether this methodology adds something novel/significant which could not be achieved by cleaning up input signal with methods that are also part of the current methodology. The authors compare against data-centric selection methods, but not against data-centric denoising methods. To this regards, I missed discussion of non-ML signal-processing work, or perhaps motivation as to why such a sophisticated/intricate framework is needed when the spectral filtering itself already appears highly effective (Figure 3). The diminishing returns on modern architectures (~0.5–1%) raise questions about long-term relevance.

**Originality**

The jump from spectral denoising + standard dropout, which are already established techniques, to the full DropoutTS pipeline isn't huge, and without an ablation comparing against just feeding the denoised signal as input, it's hard to judge how much of the gain comes from the adaptive machinery

*In conclusion*

The paper is technically solid and well-presented, with extensive experiments demonstrating consistent (if sometimes modest) gains. The main concern is whether the full adaptive dropout machinery is justified over simpler denoising baselines that the paper's own spectral analysis suggests would be effective. A direct comparison against input-level spectral denoising would strengthen the contribution.

---

> ### Author Rebuttal · Authors · 2026-03-30
>
> We thank the reviewer for the positive recommendation and for highlighting the soundness of our experimental methodology and presentation.
>
> > **Q1: Why not use the spectrally denoised signal directly as training data instead of adaptive dropout?**
>
> **A1:** Direct input-level spectral denoising uses a binary keep-or-drop strategy. While it removes noise, it irreparably damages valid rare signals often misclassified as noise. By contrast, our Capacity-Centric approach maps noise scores to dropout rates while preserving raw data exposure. Higher dropout on noisy samples limits model capacity, promoting robust generalization and avoiding noise memorization.
>
> To validate this, we conducted a new experiment comparing DropoutTS against a "Hard Denoising" (HD) baseline on the Illness dataset, where HD aggressively smooths valid epidemiological spikes:
>
> | **Backbone (Avg)** | **Raw MAE** | **HD Baseline** | **Ours (DropoutTS)** | **Δ HD** | **Δ Ours** |
> | ------------------ | ----------- | --------------- | -------------------- | -------- | ---------- |
> | Informer           | 1.901       | 1.869           | **1.772**            | ↑ 1.7%   | **↑ 6.8%** |
> | PatchTST           | 1.110       | 1.162           | **1.098**            | ↓ 4.7%   | **↑ 1.1%** |
> | TimeMixer          | 1.148       | 1.197           | **1.148**            | ↓ 4.2%   | **↑ 0.5%** |
> | **Overall**        | -           | -               | -                    | ↓ 2.4%   | **↑ 2.8%** |
>
> Results show that HD alters the Raw MAE by +1.7% (Informer), -4.7% (PatchTST), and -4.2% (TimeMixer), leading to an overall 2.4% degradation. Hard denoising universally harms modern architectures. In contrast, DropoutTS yields a consistent 2.8% improvement without modifying the input signal.
>
> > **Q2: What happens with non-stationary frequencies (e.g., chirp signals) where energy is spread across a wide band?**
>
> **A2:** As shown in Appendix C.3 and Figure 10, our toy experiments verify that fixed-threshold reconstruction already works well on chirp signals. For comprehensive evaluation, our Synth-12 benchmark includes complex patterns with diverse dynamics and noise (Appendix A). Using the Spectral Flatness Measure (SFM), DropoutTS’s dynamic soft thresholding preserves structured sparsity and time-varying frequency features without misclassifying them as noise, thus avoiding inflated noise scores and excessive dropout. We further validate its effectiveness on real-world non-stationary datasets (PhysioNet ECG, Exchange Rate), where it achieves consistent improvements; detailed results are in A4/Q4 of Reviewer AZH9.
>
> > **Q3: Why is there a 0.0% improvement on ETTm2? Does the method disengage?**
>
> **A3:** You're right that the method automatically "disengages" (defaults to standard fixed dropout) when intrinsic noise variance is uniformly low. We extracted the distribution of sample-level adaptive dropout rates ($p_i$) during the final epochs:
>
> | **Dataset** | **Mean Rate** | **Std Dev** | **Variance** | **Interpretation**  |
> | ----------- | ------------- | ----------- | ------------ | ------------------- |
> | ETTm2       | 0.329         | 0.070       | **0.004887** | Highly concentrated |
> | ILI         | 0.267         | 0.139       | **0.019436** | Widely dispersed    |
>
> On ETTm2, dropout rates are highly concentrated. As shown in Figure 12 (Row 4), ETTm2 has a dense, highly stationary oscillation pattern. With little variation in sample difficulty, our module effectively reduces to a uniform baseline dropout. In contrast, ILI has highly irregular and heterogeneous samples, leading to a dropout rate variance four times that of ETTm2. The 0.0% gain on ETTm2 validates DropoutTS’s self‑regulation: it dynamically adjusts capacity for heterogeneous data while reverting to the baseline for uniform data.
>
> > **Q4: Given diminishing returns on modern robust architectures (~0.5-1% gain), is adaptive dropout still necessary?**
>
> **A4:** While modern architectures guarantee global robustness through structural design, they fail to dynamically adapt to the learning difficulty of individual samples. With only 4 extra parameters, DropoutTS outperforms strong baselines by a stable 0.5%–1.0%, effectively combining structural robustness and dynamic capacity modulation while accelerating convergence (Table 6). Notably, DropoutTS is computationally efficient: it reduces total training time by 15.8% by avoiding high-frequency noise fitting (Reviewer kRQD A1/Q1). Its consistent improvements over SOTA models, with zero inference cost and shorter training time, are thus well justified.
>
> > **Q5: Presentation issue in Figure 4 and lack of discussion on failure modes.**
>
> **A5:** Thank you for pointing out the typo in Figure 4; the $x_{detrend}$ and $x_{trend}$ labels were swapped and are now corrected. We have added a "Limitations" section discussing diagnostic criteria for when spectral sparsity breaks down (see A1/Q1 of Reviewer NUL7).

---

> > ### Author Rebuttal · Reviewer_fGRU · 2026-04-03
> >
> > I thank the author for their extensive engagement with my concerns and questions. I appreciate the new experiments which answer some of my questions.

---

### Official Review · Reviewer_kRQD · 2026-03-13

**Soundness:** 3
**Presentation:** 4
**Significance:** 3
**Originality:** 4
**Overall Recommendation:** 5
**Confidence:** 4

**Summary:**

This paper proposes a capacity-centric regularization approach for time-series data, which leverages spectral sparsity to efficiently quantify instance-level noise through reconstruction residuals. Based on the estimated noise level, the method applies instance-adaptive Dropout.

The proposed approach, DropoutTS, functions as a model-agnostic plug-in that can be integrated into various neural network architectures. Experimental results demonstrate consistent performance improvements across multiple models and datasets.

**Compliance With Llm Reviewing Policy:**

Affirmed.

**Final Justification:**

The authors have addressed most of my concerns.

**Key Questions For Authors:**

1. Since SNS already identifies noise in the frequency domain, why is Dropout, a relatively coarse-grained regularization strategy, chosen as the main mechanism? Are there other possible regularization strategies that could be explored?

2. How does DropoutTS perform on models that already include explicit denoising or noise-robust mechanisms?

**Limitations:**

Yes. The limitations of the work are discussed in the paper.

**Strengths And Weaknesses:**

### Strengths

1. The paper introduces a capacity-centric paradigm, which evaluates samples at the instance level and applies adaptive regularization accordingly. This provides a new perspective for handling noise in time-series data.

2. DropoutTS is orthogonal to the underlying model architecture and can be easily integrated into existing models. It improves model performance with minimal additional computational overhead.

3. The method is evaluated across multiple models and datasets, and no performance degradation is observed in any configuration. This suggests that the proposed method is robust and stable.

4. The approach leverages spectral sparsity analysis, a classical signal processing technique, and quantifies noise using reconstruction error, which provides theoretical interpretability.

---

### Weaknesses

Despite its strengths, the paper still has several limitations:

1. The authors propose a capacity-centric paradigm as an alternative to traditional data-centric and prior-centric approaches. However, the paper does not provide a direct comparison with representative existing methods in terms of both performance and computational cost.

2. The Spectral Noise Scorer (SNS) relies on FFT transformation. FFT may become unstable when applied to short segments of sequences. The paper lacks an analysis of the stability of SNS under different sequence lengths, which limits the reliability of the method for short-window tasks.

---

> ### Author Rebuttal · Authors · 2026-03-30
>
> We thank the reviewer for recognizing the novelty of our capacity-centric paradigm and the model-agnostic design of DropoutTS. Below we address your specific questions.
>
> > **Q1: Lack of comparison with existing data- and prior-centric methods in terms of performance and cost.**
>
> **A1:** DropoutTS is orthogonal to existing methods and designed to complement rather than compete with them.
>
> - **Data-Centric Methods:** As shown in Table 7, DropoutTS and Selective Learning (SL) each significantly outperform baselines. Their combination achieves the best performance, verifying that DropoutTS provides complementary gains to data-centric strategies such as SL.
> - **Prior-Centric Methods:** Methods like BayesTSF involve complex architectural and loss modifications. Their non-plug-and-play nature makes a fair, mathematically consistent comparison on an identical deterministic backbone infeasible.
> - **Computational Cost:** Table 6 demonstrates that DropoutTS accelerates convergence. To explicitly address your concern, we provide a detailed comparison of parameter overhead and training time against SL (Informer backbone, Illness dataset):
>
> | **Method**         | **Base Params** | **Additional Params** | **Total Params** | **Overhead**  |
> | ------------------ | --------------- | --------------------- | ---------------- | ------------- |
> | DropoutTS (Ours)   | 11,799,047      | **+4**                | 11,799,051       | **0.000034%** |
> | Selective Learning | 11,799,047      | +3,000                | 11,802,047       | 0.025%        |
>
> DropoutTS adds only 4 parameters. Conversely, SL relies on third-party proxy models for scoring; even using the most lightweight model (DLinear), SL adds 3,000 parameters and requires a mandatory pre-training phase.
>
> | **Method**     | **Training Time** | **Pre-training Time** | **Total Time** | **vs Baseline** |
> | -------------- | ----------------- | --------------------- | -------------- | --------------- |
> | Raw Baseline   | 20.0s             | 0s                    | 20.0s          | -               |
> | DropoutTS      | **16.8s**         | **0s**                | **16.8s**      | **↓ 15.8%**     |
> | SL             | 15.1s             | 29.0s                 | 44.1s          | ↑ 121.0%        |
> | DropoutTS + SL | 6.5s              | 29.0s                 | 35.5s          | ↑ 77.5%         |
>
> Overall, DropoutTS is highly time-efficient. While SL accelerates training by discarding 20% of data, DropoutTS accelerates convergence by dynamically controlling capacity, entirely avoiding information loss.
>
> > **Q2: The Spectral Noise Scorer (SNS) relies on FFT, which may be unstable on short sequences.**
>
> **A2:** The Illness (ILI)  dataset is a typical short-sequence task (look-back 24, \(H\in\{24,36,48,60\}\); stability analysis in Appendices D and E). To reduce FFT instability, we use an adaptive amplitude threshold for low-energy noise filtering, calibrated via the Spectral Flatness Measure (SFM) to quantify signal purity, ensuring stable filtering even with small FFT windows. Figure 13 shows a strong inverse correlation between SFM and SNR, offering a reliable prior for different sequence lengths. Furthermore, to avoid spectral leakage from FFT periodicity assumptions, we apply Global Linear Detrending before transformation (Figure 14).
>
> > **Q3: Since SNS identifies noise in the frequency domain, why choose Dropout as the main mechanism?**
>
> **A3:** Dropout was chosen for three core reasons:
>
> 1. **Model-Agnostic Compatibility:** Operating directly at the feature level, DropoutTS can be easily integrated without modifying task-specific data preprocessing pipelines.
> 2. **Complementarity to Denoising:** SNS quantifies noise without applying hard input-level denoising, while DropoutTS acts on internal representations. The two methods are fully orthogonal.
> 3. **Information Preservation:** Unlike rigid sample selection or heavy input denoising, adaptive dropout allows the model to process noisy samples while limiting learning capacity. It prevents memorization of high-frequency artifacts while retaining exposure to valid rare signal patterns.
>
> We agree that exploring more fine-grained regularization is promising, and we will discuss this in our future work section.
>
> > **Q4: How does DropoutTS perform on models with explicit denoising mechanisms?**
>
> **A4:** Several baselines already integrate strong implicit or explicit noise-robust designs. For instance, TimesNet employs FFT to extract dominant periods (serving as a spectral low-pass filter), PatchTST uses patch-based aggregation (a local smoothing filter), and TimeMixer adopts multi-scale decomposable mixing.
>
> Yet, DropoutTS achieves consistent gains (Tables 2/4) due to its orthogonality: existing structural mechanisms apply uniform operations to all samples, while DropoutTS modulates functional capacity per sample, dynamically targeting residual noise that evades these filters and throttling capacity for noisy samples to avoid overfitting.

---

> > ### Author Rebuttal · Reviewer_kRQD · 2026-04-03
> >
> > Could you provide further evidence of DropoutTS's effectiveness by incorporating more recent baselines? The current selection of baselines is somewhat dated, and the performance gains over more modern models appear limited. Given that contemporary time series forecasting models now employ various advanced techniques for noise reduction, it is essential to evaluate the performance against these newer benchmarks.

---

> > > ### Author Response · Authors · 2026-04-04
> > >
> > > Thanks for your suggestion. To fully address your concerns, we evaluated DropoutTS on three 2025 models and added experiments in few-shot and zero-shot settings. Our results show that DropoutTS provides consistent, complementary gains even to the most advanced models and significantly enhances data-scarce and OOD robustness.
> > >
> > > **1. Evaluation on Recent 2025 Models**
> > >
> > > We evaluate DropoutTS on three 2025 SOTA baselines—TimeFilter (ICML 2025), WPMixer (AAAI 2025), and MultiPatchFormer (Sci. Rep. 2025)—across three representative datasets. Results are reported as MAE/MSE.
> > >
> > > | **Dataset** | **WPMixer → +DT** | **TimeFilter → +DT** | **MPFormer → +DT** |
> > > | --- | --- | --- | --- |
> > > | **Illness** | 1.065/3.081 → **1.038/2.977** | 0.965/2.407 → **0.932/2.223** | **0.991**/2.804 → 1.001/**2.550** |
> > > | **ETTh1** | 0.432/0.455 → **0.431/0.454** | 0.428/0.452 → **0.427/0.449** | 0.429/0.441 → **0.429**/**0.439** |
> > > | **Exchange Rate** | 0.449/0.445 → **0.445/0.441** | 0.452/0.446 → **0.450/0.445** | 0.406/0.357 → **0.401**/**0.351** |
> > >
> > > DropoutTS consistently reduces average MSE across all backbones on the heterogeneous Illness dataset (e.g., MPFormer: 2.804 to 2.550; TimeFilter: 2.407 to 2.223). This suggests that instance-level modulation effectively mitigates the local noise that often misleads global structural filters. Furthermore, DropoutTS achieves consistent gains on ETTh1 and Exchange Rate, demonstrating its seamless, non-interfering integration even when strong periodicity or non-stationarity pushes baselines to their limits.
> > >
> > > **2. Few-Shot Robustness: Preventing Memorization**
> > >
> > > Interestingly, we found that the advantages of DropoutTS become even more pronounced in data-scarce scenarios. To illustrate this, we evaluated PatchTST on the Illness dataset using varying fractions of training data.
> > >
> > > | **Training Data Fraction** | **PatchTST** | **+ DropoutTS** | **MAE Improv. (Δ)** |
> > > | --- | --- | --- | --- |
> > > | 10% (0.1p) | 1.333 / 4.185 | 1.295 / 4.074 | **+2.85% ↑** |
> > > | 25% (0.25p) | 1.156 / 3.201 | 1.144 / 3.195 | **+1.04% ↑** |
> > > | 50% (0.5p) | 1.116 / 3.143 | 1.106 / 3.169 | **+0.90% ↑** |
> > > | 75% (0.75p) | 1.111 / 3.169 | 1.096 / 3.128 | **+1.35% ↑** |
> > > | 100% (1.0p) | 1.110 / 3.322 | 1.098 / 3.126 | **+1.08% ↑** |
> > >
> > > The largest gain (+2.85% MAE) appears at 10% data, where models are most prone to memorizing noise with low sample diversity. DropoutTS mitigates this by dynamically constraining functional capacity.
> > >
> > > Notably, the baseline trained on just 75% of the data matches or exceeds the 100% model. This reveals significant data redundancy in the dataset—raw data scaling offers diminishing returns. The next breakthrough in forecasting will likely come from data-centric innovations rather than architectural scaling.
> > >
> > > **3. Zero-Shot OOD Generalization**
> > >
> > > Similarly, we observed that DropoutTS significantly enhances model robustness in zero-shot Out-Of-Distribution (OOD) scenarios. We validated this by transferring models across ETT datasets with varying sampling frequencies and evolution patterns.
> > >
> > > | **Zero-Shot Transfer** | **PatchTST** | **+ DropoutTS** |
> > > | --- | --- | --- |
> > > | ETTh1 $\rightarrow$ ETTh2 | 0.391 / 0.406 | **0.387** / **0.404** |
> > > | ETTh1 $\rightarrow$ ETTm2 | 0.319 / 0.359 | **0.318** / **0.358** |
> > > | ETTh2 $\rightarrow$ ETTh1 | 0.519 / 0.480 | **0.501** / **0.473** |
> > > | ETTh2 $\rightarrow$ ETTm2 | 0.325 / 0.364 | **0.317** / **0.358** |
> > > | ETTm1 $\rightarrow$ ETTh2 | 0.457 / 0.442 | **0.448** / **0.439** |
> > > | ETTm1 $\rightarrow$ ETTm2 | 0.305 / 0.331 | **0.302** / **0.330** |
> > > | ETTm2 $\rightarrow$ ETTh2 | 0.424 / 0.428 | **0.420** / **0.427** |
> > > | ETTm2 $\rightarrow$ ETTm1 | 0.476 / 0.444 | **0.470** / **0.442** |
> > >
> > > Baseline models overfit source noise and struggle under distribution shifts, while DropoutTS improves robustness by filtering fluctuations and learning invariant dynamics.
> > >
> > > **Concluding Remarks: A Data-Centric Future**
> > >
> > > As documented by the Accuracy Law [1], purely architectural innovations in time series forecasting are rapidly saturating. With year-over-year gains on benchmark dropping from 14.98% (2022) to 3.51% (2025), chasing marginal architectural improvements is yielding diminishing returns.
> > >
> > > This stagnation exposes a fundamental bottleneck: while time series data is highly heterogeneous, modern architectures continually impose rigid structural biases. DropoutTS shifts the paradigm from *model-centric design* to *data-centric dynamic adaptation*. By evaluating heavily utilized, stable touchstones (e.g., PatchTST, TimeMixer) alongside the new 2025 SOTAs, we consistently demonstrate that instance-level capacity modulation is a more rigorous and promising path forward than rigid structural filters.
> > >
> > > We envision dynamic capacity modulation not merely as an optional plugin, but as a built-in necessity for future architectures.
> > >
> > > We hope this perspective inspires new work along these lines.
> > >
> > > [1] Wang, Y. et al. "Accuracy law for the future of deep time series forecasting." arXiv:2510.02729 (2025).

---

### Decision · Program_Chairs · 2026-04-30

**Decision:**

Accept (regular)

**Comment:**

This paper proposes DropoutTS, a novel regularization technique for time-series data. It introduces a capacity-centric modeling paradigm, where a noisy time-series signal undergoes spectral analysis to estimate instance-wise noise levels, which are then used to adapt dropout rates—assigning higher dropout to noisier samples. The proposed approach is model-agnostic and orthogonal to existing regularization methods, making it easy to integrate with a wide range of architectures.

All reviewers find the contribution to be novel, interesting, and technically sound. While there were initial concerns regarding missing ablations, comparisons to stronger baselines, and empirical validation in broader settings, the authors addressed these issues during the rebuttal, which strengthened the overall evaluation. Some reviewers note that the empirical gains are more modest on recent state-of-the-art models and raise questions about the necessity of the full adaptive framework over simpler alternatives, but these concerns do not outweigh the overall contribution.